# VersVideo: Leveraging Enhanced Temporal Diffusion Models for Versatile Video Generation

**Jinxi Xiang**[1], **Ricong Huang**[2], **Jun Zhang**[1*], **Guanbin Li**[2], **Xiao Han**[1], **Yang Wei**[1]

[1]Tencent AI Lab, Shenzhen, China

[2]School of Computer Science and Engineering, Sun Yat-sen University, Guangzhou, China

* Corresponding author: junejzhang@tencent.com

## Abstract

Creating stable, controllable videos is a complex task due to the need for significant variation in temporal dynamics and cross-frame temporal consistency. To address this, we enhance the spatial-temporal capability and introduce a versatile video generation model, `VersVideo`, which leverages textual, visual, and stylistic conditions. Current video diffusion models typically extend image diffusion architectures by supplementing 2D operations (such as convolutions and attentions) with temporal operations. While this approach is efficient, it often restricts spatial-temporal performance due to the oversimplification of standard 3D operations. To counter this, we incorporate two key elements: (1) multi-excitation paths for spatial-temporal convolutions with dimension pooling across different axes, and (2) multi-expert spatial-temporal attention blocks. These enhancements boost the model's spatial-temporal performance without significantly escalating training and inference costs. We also tackle the issue of information loss that arises when a variational autoencoder is used to transform pixel space into latent features and then back into pixel frames. To mitigate this, we incorporate temporal modules into the decoder to maintain inter-frame consistency. Lastly, by utilizing the innovative denoising UNet and decoder, we develop a unified ControlNet model suitable for various conditions, including image, Canny, HED, depth, and style. Examples of the videos generated by our model can be found at `https://jinxixiang.github.io/versvideo/`.

## 1 Introduction

Diffusion models have transformed automated content creation, empowering designers to generate highly realistic images or videos from textual prompts as inputs (Ho et al., 2022a;b; Rombach et al., 2021; Singer et al., 2022; Huang et al., 2024). However, despite significant strides in image generation, the quality of generated videos often fails to match real-world footage. This discrepancy can be attributed to the inherent high dimensionality and complexity of videos, which encapsulate intricate spatiotemporal dynamics within high-resolution frames. In this paper, we leverage diffusion models for video generation, specifically addressing three key challenges: (i) the fundamental building block in UNet for handling spatial-temporal dynamics; (ii) the issues of a variation autoencoder to transform a video into latent and subsequently reconstruct it back into pixel space, and (iii) the capacity to control video generation under diverse conditions.

**UNet architecture.** The creation of open-domain text-to-video models is widely acknowledged as a significant challenge, primarily due to the scarcity of large-scale text-video paired data and the complexity of constructing space-time models from the ground up. To mitigate these challenges, most current methodologies are built upon pre-trained image generation models (Ho et al., 2022b; Wu et al., 2022; Chen et al., 2023a; Xing et al., 2023a). These methodologies typically adopt space-time separable architectures, where spatial operations are derived from the image generation model. To incorporate temporal modeling, various strategies have been employed, including pseudo-3D modules (Singer et al., 2022; Zhou et al., 2022), serial 2D and 1D blocks (Saharia et al., 2022; Xing et al., 2023a), and parameter-free techniques such as temporal shift (An et al., 2023a; Zhang et al.,

2023) or custom spatiotemporal attention (Wu et al., 2022; Wang et al., 2023a). However, these methodologies often overlook the critical interaction between time and space, which is essential for creating visually engaging text-to-video content.

Specifically, parameter-free approaches (An et al., 2023b; Wu et al., 2022) rely on manually designed rules that often fail to capture the intrinsic nature of videos, resulting in the generation of unnatural motions. Learnable 2D+1D modules and blocks (Blattmann et al., 2023; Singer et al., 2022; Xing et al., 2023a) primarily concentrate on temporal modeling, either by directly feeding temporal features to spatial features or combining them through simplistic element-wise additions. This limited interaction often results in temporal distortions and mismatches between the input texts and the generated videos, adversely affecting the overall quality of the generated content. Recent advancements in convolution designs for video classification (Wang et al., 2021; Li et al., 2020; Jiang et al., 2019; Wang et al., 2021; Liu et al., 2023a), as well as attention mechanisms for vision transformers (Puigcerver et al., 2023; Huang et al., 2023b), have demonstrated significant potential for efficient and robust spatial-temporal modeling. *We aim to address these challenges by advancing our approach with a highly efficient design of spatial-temporal convolution and attention blocks.*

**Autoencoder.** Instead of training the model on raw pixels, the Latent Diffusion Model (LDM) first uses an autoencoder to learn a low-dimensional latent space that succinctly parameterizes images, and then models this latent distribution. Building on this concept, numerous video diffusion models (Blattmann et al., 2023; An et al., 2023a; Wang et al., 2023d; He et al., 2022b; Liu et al., 2023a) also employ an autoencoder to project videos into latent space, significantly reducing the computational load when training on high-resolution videos. To leverage the pre-trained image diffusion model, many of these models (Liu et al., 2023a) repurpose the image autoencoder, focusing primarily on the UNet model design. However, while these models have demonstrated promising results in image reconstruction, the time-independent image autoencoder can distort temporal information in the latent space (Zhou et al., 2022). The pre-trained image autoencoder has also been reported to cause image distortions due to the high compression rate (Lin et al., 2023; Zhu et al., 2023). Lin et al. (2023) suggests using frequency-compensated modules to alleviate this issue. *To address these challenges, we enhance the spatial-temporal reconstruction capability of the autoencoder by training a new decoder to counteract information loss that occurs during projection and reconstruction with spectrum constraints.*

**Controllable generation.** The ability to control generation is a highly sought-after feature in generative foundation models. In the realm of image generation, models such as ControlNet (Zhang & Agrawala, 2023), T2I Adapter (Mou et al., 2023), and Composer (Huang et al., 2023a) have been developed to manage visual conditions like edge maps and depth maps. When it comes to video generation, VideoComposer (Wang et al., 2023c) enhances video synthesis by providing superior control over both spatial and temporal dimensions. This is accomplished by breaking down a video into three primary factors: textual conditions, spatial conditions, and crucial temporal conditions. A latent diffusion model then uses these conditions to reconstruct the input video. Another method, similar to the image ControlNet, is Control-A-Video (Chen et al., 2023b). This model also has the ability to handle visual conditions. However, each ControlNet model is restricted to managing a specific control modality it was trained on, such as an edge map. To handle a different visual condition modality, it becomes necessary to train a separate model, leading to significant time and spatial complexity costs. *Inspired by the Uni-ControlNet (Zhao et al., 2023) used in image generation, we propose a unified and scalable video control model. This model can be smoothly integrated with our diffusion foundation model, resulting in a less complex training process.*

In summary, we present a versatile video generation model that incorporates innovative concepts in (i) The development of a novel UNet video diffusion model: By combining more robust multi-excitation spatial-temporal convolution and Mixture of Experts attention blocks, we enhance the model without significantly increasing training and inference complexity. (ii) The implementation of a spatial-temporal compensated decoder for video reconstruction: We design a VAE decoder with temporal modules and utilize a refinement network to improve reconstruction quality. (iii) The establishment of a unified control for video generation: By extending the proposed diffusion model with a shared ControlNet and diverse adapters, we achieve comprehensive video control generation.

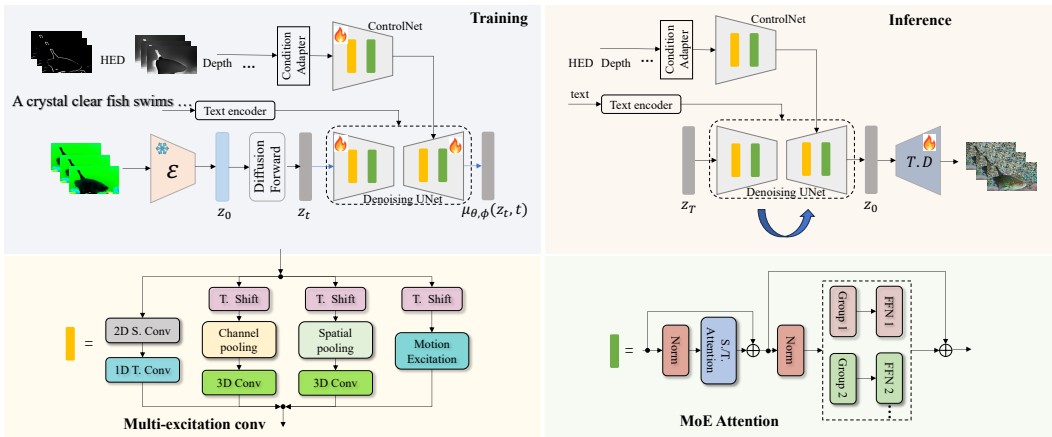

Figure 1: The `VersVideo` builds on multi-excitation convolution and MoEs spatial-temporal attention blocks. Unified condition adapters and a shared ControlNet are used for controllable generation. A temporal compensated video decoder addresses potential inconsistencies during video reconstruction. 'T.' and 'S.' denote temporal and spatial, respectively.

## 2 METHODS

### 2.1 DIFFUSION PROBABILISTIC MODELS

DPM, first introduced in (Sohl-Dickstein et al., 2015), is a powerful generative model that has been successfully applied to various tasks, such as image synthesis and video generation. It consists of a diffusion process and a denoising process. In the diffusion process, random noises are gradually added to the data $x$ via a T-step Markov chain (Kong & Ping, 2021). The noised latent variable at step $t$ can be expressed as:

$$\mathbf{z}_t = \sqrt{\hat{\alpha}_t}x + \sqrt{1 - \hat{\alpha}_t}\epsilon_{\mathbf{t}}, \text{with } \hat{\alpha}_t = \prod_{k=1}^{t}\alpha_k \quad \epsilon_{\mathbf{t}} \sim \mathcal{N}(\mathbf{0}, \mathbf{1}), \tag{1}$$

where $\alpha_t \in (0, 1)$ is the corresponding coefficient. For a $T$ that is large enough, e.g., $T = 1000$, we have $\sqrt{\hat{\alpha}_T} \approx 0$ and $\sqrt{1 - \hat{\alpha}_T} \approx 1$. And $\mathbf{z}_T$ approximates a random Gaussian noise. Then the generation of $\mathbf{x}$ can be modeled as iterative denoising.

Ho et al. (2020) connect DPM with denoising score matching and propose a $\epsilon-$prediction form for the denoising process:

$$\mathcal{L}_t = \|\epsilon_t - \mathbf{z}_\theta\left(\mathbf{z}_t, t\right)\|^2, \tag{2}$$

where $\mathbf{z}_\theta$ is a denoising neural network parameterized by $\theta$, and $\mathcal{L}_t$ is the loss function.

### 2.2 DENOISING UNET

As shown in Fig. 1, The diffusion UNet employed in `VersVideo` is built upon Multi-Excitation Convolution (MEC) and Mixture-of-Experts Spatial-Temporal Attention (MoE-STA) blocks. A temporal-compensated decoder is utilized to reconstruct videos from the latent space back to the pixel domain. By incorporating an auxiliary ControlNet, `VersVideo` is capable of handling various input conditions, such as text, images, HED, depth, and Canny maps.

**Multi-Excitation Convolution.** In contrast to existing video diffusion models that often limit spatial-temporal performance due to over-simplification of standard 3D operations, we introduce multi-excitation paths for spatial-temporal convolutions with dimension pooling across different axes. Our design is fundamentally inspired by squeeze-and-excitation (SE) block (Hu et al., 2017; Li et al., 2020) by explicitly modeling channel/temporal dependencies. The excitation paths consist of (i) factorized 2D spatial + 1D temporal convolution; (ii) spatial-temporal excitation; (iii) channel excitation; and (iv) motion excitation.

Given a batch of video latents $\mathbf{z} \in \mathbb{R}^{B \times C \times T \times H \times W}$. Notations used in this section are $B$ (batch size), $C$ (channels), $T$ (number of frames), $H$ (height), and $W$ (width). The factorized 2D + 1D branch first reshapes $\mathbf{z}$ into $(BT, C, H, W)$ for spatial convolution, and then processes it with temporal pseudo-3D kernel (3, 1, 1) in the shape of $(B, C, T, H, W)$.

For the other three branches, the input tensors $\mathbf{z}$ are shifted temporally to $\hat{\mathbf{z}}$ following TSM (Lin et al., 2018) before processing, which is computationally free but has strong spatiotemporal modeling ability. The motivation behind this is that both the channel and spatial axes have high dimensions, driving us to pool over these axes and apply convolution separately. For spatial-temporal excitation, we average $\hat{\mathbf{z}}$ along channel axis into $\mathbf{F} \in \mathbb{R}^{B \times 1 \times T \times H \times W}$. We apply 3D convolution to $\mathbf{F}$ to get the excitation mask $\mathbf{M}_{\mathrm{S}}$. The output of this branch is:

$$\mathbf{z}_{\mathrm{SPE}} = \hat{\mathbf{z}} + \hat{\mathbf{z}} \odot \mathbf{M}_{\mathrm{S}} = \hat{\mathbf{z}} + \hat{\mathbf{z}} \odot \mathrm{sigmoid}\left(\mathrm{Conv}_{\mathrm{3D}}\left(\mathbf{F}\right)\right). \tag{3}$$

For channel excitation, we reduce $\hat{\mathbf{z}}$ through spatial averaging into $\mathbf{C}_1 \in \mathbb{R}^{B \times C \times T \times 1 \times 1}$, and then squeezing the channels into $\mathbf{C}_2 \in \mathbb{R}^{B \times C/16 \times T \times 1 \times 1}$ through $1 \times 1$ convolution (the channel and temporal should swap for convolution and revert back). Pseudo 3D convolution $(3, 1, 1)$ is applied for temporal reasoning, after that we extend the channel back to the original size and get $\mathbf{C}_3 \in \mathbb{R}^{B \times C \times T \times 1 \times 1}$. Using $\mathbf{C}_3$ as the excitation, we gate the $\hat{\mathbf{z}}$ with the excitation mask $\mathbf{M}_C$ to get:

$$\mathbf{z}_{\mathrm{C}} = \hat{\mathbf{z}} + \hat{\mathbf{z}} \odot \mathbf{M}_{\mathrm{C}} = \hat{\mathbf{z}} + \hat{\mathbf{z}} \odot \mathrm{sigmoid}\left(\mathbf{C}_3\right). \tag{4}$$

Motion excitation has been explored in previous studies (Li et al., 2020; Jiang et al., 2019; Wang et al., 2021; Liu et al., 2023a), as shown in Fig. 2. The motion information is extracted by differentiating the adjacent frames. The input frames are squeezed along the channel axis and the motion of $\hat{\mathbf{z}}_{t-1}$ and $\hat{\mathbf{z}}_t$ is expressed as:

$$\mathbf{m}_{t-1} = \mathbf{K}_{\mathrm{2D}} * \hat{\mathbf{z}}_t - \hat{\mathbf{z}}_{t-1}. \tag{5}$$

Motion results $\{\mathbf{m}_1, ... \mathbf{m}_T\}$ are stacked along the temporal axis to reconstruct the video tensor. Then, the tensor is pooled on the spatial space and unsqueeze along the channel axis. Finally, the motion tensor is activated with sigmoid to modulate $\hat{\mathbf{z}}$.

**MoEs Spatial-Temporal Attention** The UNet incorporates a series of convolutional blocks with attention blocks. The 3D attention has a prohibitive complexity of $O((HWT)^2)$ for tensor $(B, HWT, d)$, where $HW$ denotes the spatial scale, $T$ is the number of frames, and $d$ represents the feature dimension.

We use factorized attention to conduct **spatial attention** in the shape of $(BT, HW, d)$ and subsequently apply **temporal attention** $(BHW, T, d)$. To enhance this attention, we augment it with MoE design without significant increases in computational cost. In specific, we create MoE layers with $N$ spatial experts (i.e., $N$ FFNs) $\{E_{S,1}, E_{S,2}, ..., E_{S,N}\}$ and $M$ temporal experts $\{E_{T,1}, E_{T,2}, ..., E_{T,M}\}$ (Huang et al., 2023b). Without loss of generality, we denote input tokens as $(B, L, d)$ where $L = HW$ for spatial attention and $L = T$ for temporal attention. Assuming $L$ is divisible by $N$ (or $M$), we randomly split the tokens into $N$ (or $M$) groups and then process them with experts:

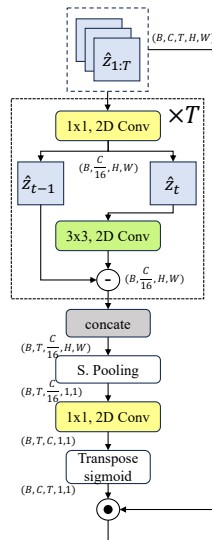

Figure 2: Motion Excitation.

$$\{x_1, x_2, ..., x_L\} \xrightarrow{\text{shuffle}} \{\hat{x}_1, \hat{x}_2, ..., \hat{x}_L\} \xrightarrow{\text{split}} \{X_1, X_2, ..., X_N\} \xrightarrow{\text{feedforward}} Y_i = E_{S,i}(X_i), \tag{6}$$

where $X_1 = \{\hat{x}_1, \hat{x}_2, ..., \hat{x}_{L/N}\}$, $X_2 = \{\hat{x}_{L/N+1}, \hat{x}_2, ..., \hat{x}_{2L/N}\}$, etc. We unshuffle $\{Y_1, Y_2, ..., Y_N\}$ to recover the output of the MoE layer. The temporal MoE layer follows the same operation.

After each training iteration, we update experts by averaging all of them:

$$\overline{E}_{S,i} = (1-\lambda)E_{S,i} + \lambda \overline{E}_{S,:} \quad \overline{E}_{T,i} = (1-\lambda)E_{T,i} + \lambda \overline{E}_{T,:}, \tag{7}$$

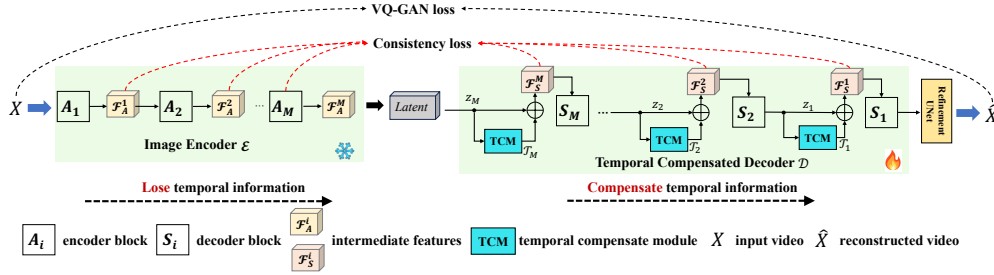

Figure 3: The proposed TCM-VAE. Many current video diffusion models rely on image VAEs for video encoding and reconstruction, which often neglects crucial temporal information (Wu et al., 2022; Wang et al., 2023d). The proposed TCM-VAE builds on image VAE by introducing TCMs and a refinement UNet to ensure temporal consistency and video quality.

with:

$$\overline{E}_{S,:} = \Sigma_{j \neq i}^{N} \frac{1}{N-1} E_{S,j} \quad \overline{E}_{T,:} = \Sigma_{j \neq i}^{M} \frac{1}{M-1} E_{T,j}, \tag{8}$$

where $\overline{E}_{S,i}$ and $\overline{E}_{T,i}$ the $i$-th spatial/temporal expert; $\lambda$ controls the information exchange rate among experts. Since a large dropout $\frac{N-1}{N}$ (or $\frac{M-1}{M}$) is applied for every expert, we have to communicate the weights among all of them to avoid data insufficiency and underfit problem. By default, we set $\lambda = 0.001$ as a small value, ensuring experts gradually evolve together, while in the meantime, avoiding weights collapse where all experts are identical unexpectedly.

During inference, each MoE layer is transformed into an FFN layer by averaging $\text{FFN}_S = \frac{1}{N}\Sigma_{i=1}^{N} E_{S,i}$ and $\text{FFN}_T = \frac{1}{M}\Sigma_{i=1}^{M} E_{T,i}$. In this way, introducing MoE to enhance spatial-temporal attention does not increase inference computation cost. Merging weights of models for better performance is widely acknowledged in large language models (Jin et al., 2022) and image classification (Wortsman et al., 2022; Ainsworth et al., 2022).

## 2.3 TEMPORAL COMPENSATED DECODER

Most existing video latent diffusion models utilize image VAEs to reconstruct videos from latents (Wu et al., 2022; Chen et al., 2023b; Xing et al., 2023b; Wang et al., 2023d). However, an image VAE without temporal modules fails to capture the dependencies between frames. This leads to a loss of temporal information in videos, which in turn results in flickering artifacts and temporal inconsistency (Blattmann et al., 2023).

We introduce Temporal Compensation Modules (TCMs) to enhance temporal consistency, and a refinement UNet for quality enhancement, building upon an image pretrained VAE. To optimize training efficiency, we freeze the encoder and concentrate on training the video decoder. We incorporate TCMs into the decoder blocks to infuse temporal information into multi-level intermediate features. The implementation of TCMs leverages the multi-excitation module in Section 2.2. More details about the TCM-VAE can be found in the **Appendix**.

We fine-tune the decoder with the added TCMs using video data and incorporate a path-wise temporal discriminator constructed from 3D convolutions. The optimization loss on the reconstructed image $\hat{X}$ aligns with that of the VQ-GAN (Esser et al., 2020). For intermediate features derived from TCMs, we impose a consistency loss constraint on the feature, aligning it with features of the same level. The underlying principle is that temporal information is gradually lost during encoding; this process can be counteracted through consistency constraints. The final optimization loss for training the video decoder is formulated as follows:

$$\mathcal{L}_{\text{dec}} = \mathcal{L}_{\text{VQ-GAN}}(\mathbf{X}, \hat{\mathbf{X}}) + \lambda \Sigma_{i=1}^{M} \mathcal{L}_{\text{TCM}}(\mathcal{F}_A^i, \mathcal{F}_S^i) \tag{9}$$

where $\lambda$ is a weight coefficient ($10^{-4}$ by default); $\mathcal{L}_{\text{VQ-GAN}}$ is the reconstruction loss as in VQ-GAN, with L1, perceptual and discriminator loss; $\mathcal{L}_{\text{TCM}}$ constrains the similarity between the encoder features and TCM features, for instance, through the L2 loss.

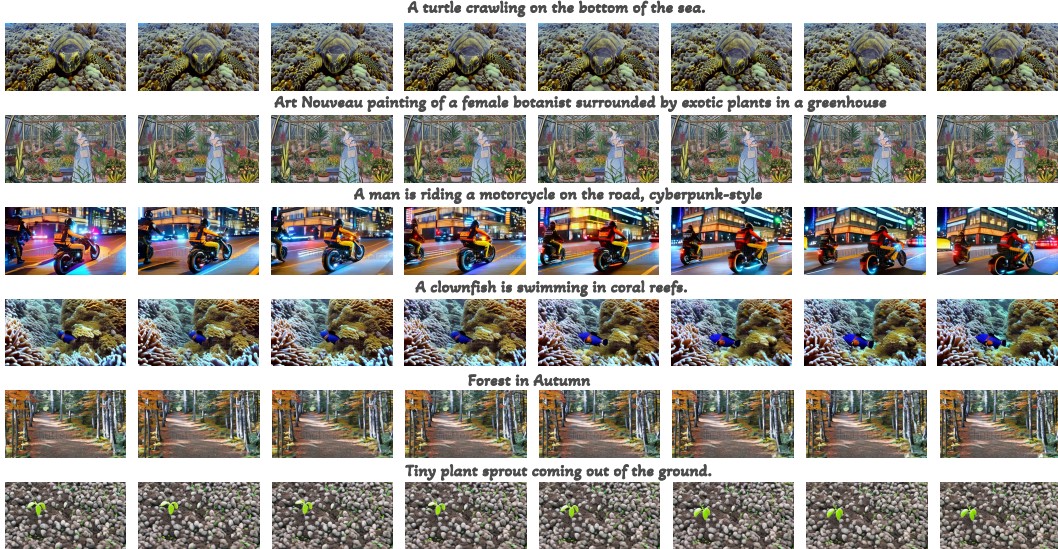

Figure 5: Sample text-to-video outputs generated by `VersVideo-H`, with a resolution of $576 \times 320$ pixels, are displayed. Frames are selected at a rate of one every four frames for analysis.

## 2.4 VERSATILE CONDITIONS GENERATION

For training versatile conditions that guide video generation, we freeze the denoising UNet and train the unified ControlNet for visual conditions. Additionally, we train the linear adapters of CLIP image for style as illustrated in Fig. 4.

The extension of the image ControlNet for video generation builds upon previous studies (Zhang et al., 2023; Chen et al., 2023b). Unlike previous models that train separate video ControlNets for specific conditions like canny, depth, HED, etc., we propose a unified approach to handle visual conditions using a shared ControlNet because diverse visual conditions share common low-level visual features. Input visual conditions are preprocessed using lightweight adapters and the shared ControlNet to introduce controllability into the UNet.

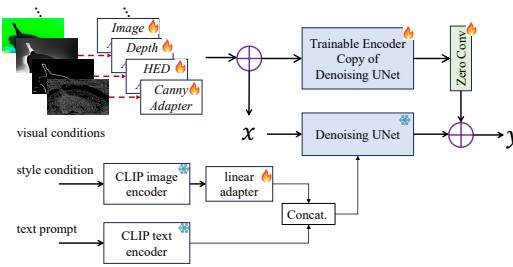

Figure 4: Versatile conditions strategy.

To adapt to various low-level visual conditions, we create adapter modules to capture features before applying the shared ControlNet. Mathematically, the adapter module can be expressed as:

$$\mathcal{F}_{\text{Adapter}}\left(\mathcal{I}_c^k\right) := \Sigma_{i=1}^{K}\mathbf{1}(i == k)\mathcal{F}_{\text{Cov}}^{(i)}\left(\mathcal{I}_c^k\right) \tag{10}$$

where $\mathbf{1}$ is the indicator function; $\mathcal{I}_c^k$ is the $k$-th visual condition; $\mathcal{F}_{\text{Cov}}^{(i)}$ is the adapter convolution.

For the style condition, which serves as global guidance similar to the text prompt, we propose training a linear adapter in conjunction with the frozen CLIP image encoder (Ye et al., 2023). Typically, the linear adapter for text prompts functions as an identity mapping. The image embeddings for style control and text embeddings are subsequently concatenated and fed into the cross-attention modules within both the denoising UNet and the ControlNet.

Table 1: Text-to-Video generation on MSR-VTT (Xu et al., 2016). ↓ denotes the lower the better. ↑ denotes the higher the better.

| Method | Training Data | FVD (↓) | CLIPSIM(↑) |
|---|---|---|---|
| GODIVA (Wu et al., 2021a) | MSR-VTT | - | 0.2402 |
| NÜWA (Wu et al., 2021b) | MSR-VTT | - | 0.2439 |
| Make-A-Video (Singer et al., 2022) | WebVid-10M + HD-VILA-10M | - | 0.3049 |
| VideoFactory (Wang et al., 2023b) | WebVid-10M + HD-VG-130M | - | 0.3005 |
| LVDM (He et al., 2022a) | WebVid-2M | 742 | 0.2381 |
| MMVG (Fu et al., 2022) | WebVid-2.5M | - | 0.2644 |
| CogVideo (Hong et al., 2022) | WebVid-5.4M | 1294 | 0.2631 |
| ED-T2V (Liu et al., 2023b) | WebVid-10M | - | 0.2763 |
| MagicVideo (Zhou et al., 2022) | WebVid-10M | 998 | - |
| Video-LDM (Blattmann et al., 2023) | WebVid-10M | - | 0.2929 |
| VideoComposer (Wang et al., 2023d) | WebVid-10M | 580 | 0.2932 |
| Latent-Shift (An et al., 2023b) | WebVid-10M | - | 0.2773 |
| VideoFusion (Luo et al., 2023) | WebVid-10M | 581 | 0.2795 |
| SimDA (Xing et al., 2023b) | WebVid-10M | 456 | 0.2945 |
| `VersVideo-L` (500M) | WebVid-10M | 620 | 0.2796 |
| `VersVideo-H` (2B) | WebVid-10M | **421** | **0.3014** |

## 3 EXPERIMENTS

### 3.1 IMPLEMENTATION DETAILS

**Training.** The training of `VersVideo` consists of two stages. In the first stage, we train the text-to-video video diffusion model using the WebVid-10M video-text dataset (Bain et al., 2021) and the 300M image-text datasets from LAION-5B (Sohl-Dickstein et al., 2015). We allocate $\frac{1}{4}$ of the GPUs for updating image-text data, while the remaining GPUs are used for video-text data. We develop `VersVideo-L`, which has 500 million parameters for $16 \times 256 \times 256$ generation, and `VersVideo-H`, which has 2 billion parameters for $24 \times 576 \times 320$ generation. *The training details and network specifications are defined in the Appendix.* In the second stage, we freeze the UNet and train the visual conditions ControlNet and style conditions using the WebVid-10M dataset.

**Inference.** Video generation combines text, visual conditions, and style are presented. We generate the video using the target prompt via DDIM sampling (Song et al., 2020).

**Evaluation.** We evaluated our model on the MSR-VTT (Xu et al., 2016) and UCF-101 datasets using the FVD (Unterthiner et al., 2018), CLIPSIM (Radford et al., 2021), and IS metrics. In the ablation study, we further employ frame consistency (FC) to assess video continuity by calculating the average CLIP similarity between two successive frames (Wang et al., 2023d; Esser et al., 2023).

### 3.2 TEXT-TO-VIDEO GENERATION

The MSR-VTT test set consists of 2,990 examples, each containing 20 descriptions or prompts. To generate videos, we randomly select one prompt from each example, resulting in 2,990 videos. Each video consists of 16 frames at a frame rate of 30 fps. For `VersVideo-H`, we extract 16 consecutive frames from a total of 24 frames. These frames are then resized and center-cropped to dimensions of $256 \times 256$. To evaluate the quality of the generated videos, we calculate the CLIPSIM score by averaging the scores of the 47,840 frames. For comparative FVD statistics, we randomly select video sequences from the dataset, ensuring that each sequence contains a minimum of 16 frames.

In Table 1, our approach achieves an average CLIPSIM score of 0.3014, even in a zero-shot scenario, outperforming the majority of competing methods. This indicates a robust semantic correlation between the generated videos and the input text. It is worth noting that Make-A-Video (Singer et al., 2022) and VideoFactory (Wang et al., 2023b) achieve higher CLIP scores, but they leverage additional large-scale HD-VILA (Xue et al., 2021) datasets during their training process.

We also provide visualizations of the generated videos for various text prompts in Fig. 5. These examples demonstrate `VersVideo`'s ability to generate diverse videos with high frame consistency.

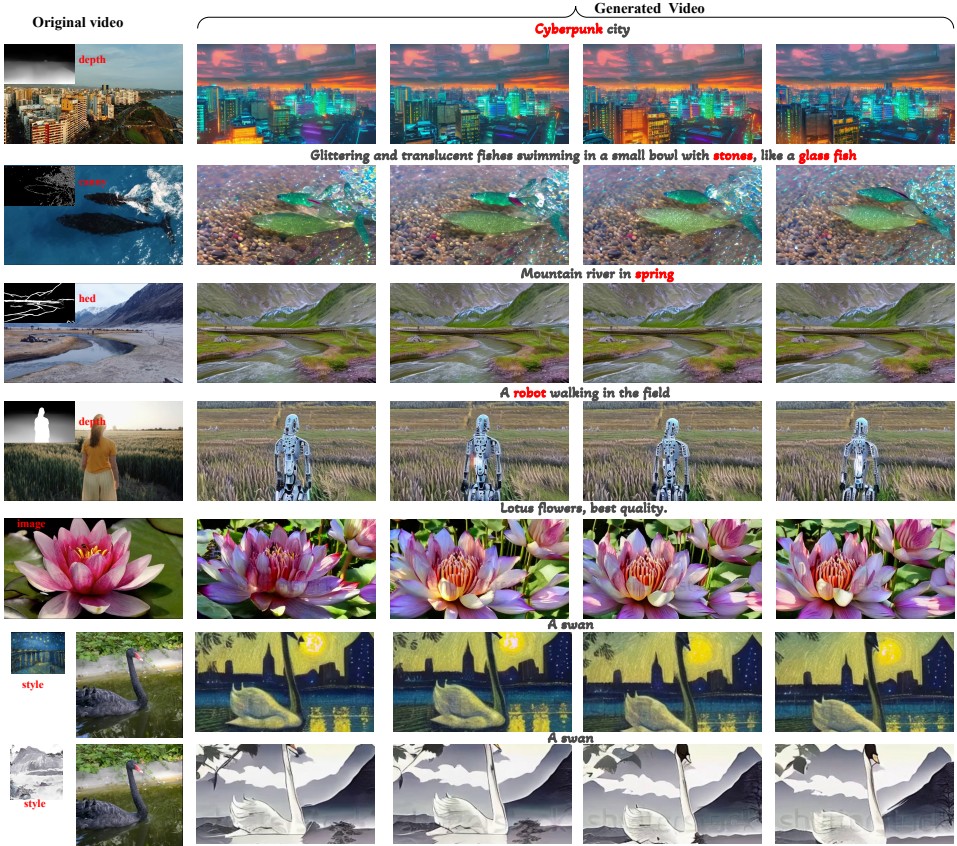

Figure 6: Conditioned video generation with image, depth, canny, HED, and style.

### 3.3 CONDITIONS GENERATION

**Visual conditions.** We categorize elements such as images, HED, Canny, depth, etc., as visual conditions, as they provide spatial information that guides the video generation process. In the `VersVideo` model, these conditions are processed using individual adapters, which then introduce controllable features into the denoising UNet. Fig. 6 showcases the results of incorporating diverse text prompts with visual conditions, demonstrating the effectiveness of our approach.

**Style condition.** The style condition serves as a global guide. It is extracted using a multi-modal CLIP image encoder and subsequently concatenated with text embeddings. The last two rows in Fig. 6 show two distinct painting styles for video generation, using depth maps as the spatial control.

### 3.4 ABLATION STUDIES

**UNet model design.** We carried out an ablation study to validate the design of the denoising UNet model, incorporating both the MEC and MoE-STA components. We trained various versions of the `VersVideo-L` model, and `VersVideo-H` and subsequently evaluated their class-conditional generation using the UCF-101 dataset. Table 2 presents the study's findings. Both the MEC and MoE-STA have demonstrated their effectiveness as design elements in improving the performance of the video diffusion model. These elements consistently contribute to performance enhancements. Although the `VersVideo-L` model does not surpass state-of-the-art models like Make-a-Video (Singer et al., 2022) and VideoFusion (Luo et al., 2023), which boast billions of parameters, it is worth noting that the `VersVideo-L` model has a significantly smaller parameter count of 500 M. Notably, the `VersVideo-H` model achieves an IS score of 81.3 and an FVD score of 119, making it the second-best performing model after Make-A-Video (9.72B parameters). The FC consistently improves with our designed modules.

Table 2: Ablation of model designs on UCF101. FC means frame consistency.

| Method | Resolution | IS ↑ | FVD ↓ | FC ↑ |
|---|---|---|---|---|
| MoCoGAN-HD (Tulyakov et al., 2017) | $128 \times 128$ | 12.42 | - | - |
| CogVideo (Hong et al., 2022) | $160 \times 160$ | 50.46 | 626 | - |
| DVD-GAN (Clark et al., 2019) | $128 \times 128$ | 32.97 | - | - |
| TATS (Ge et al., 2022) | $128 \times 128$ | 79.28 | 332 | - |
| VideoFusion (Luo et al., 2023) | $128 \times 128$ | 80.03 | 173 | - |
| Make-A-Video (Singer et al., 2022) | $256 \times 256$ | 82.55 | 81.25 | - |
| VersVideo-L (baseline) | $256 \times 256$ | 60.2 | 355 | 0.843 |
| VersVideo-L (+ MEC) | $256 \times 256$ | 68.8 | 287 | 0.862 |
| VersVideo-L (+ MoE-STA) | $256 \times 256$ | 63.5 | 302 | 0.861 |
| VersVideo-L (+ MEC + MoE-STA) | $256 \times 256$ | 72.9 | 207 | 0.880 |
| VersVideo-H | $576 \times 320$ | 81.3 | 119 | 0.905 |

Table 3: TCM-VAE for video reconstruction and generation on UCF-101.

| | Video Reconstruction | | Text-to-video Generation | | |
|---|---|---|---|---|---|
| | PSNR↑ | SSIM↑ | IS↑ | FVD↓ | FC↑ |
| vanilla image decoder | 32.376 dB | 0.9129 | 43.23 | 598 | 0.865 |
| 3D convolution decoder | 32.692 dB | 0.9181 | 49.40 | 508 | 0.868 |
| temporal-finetuned decoder | 32.254 dB | 0.9194 | 50.39 | 490 | 0.871 |
| **TCM-VAE (Ours)** | **33.019 dB** | **0.9253** | **60.18** | **443** | 0.880 |

**TCM-VAE.** To evaluate the effectiveness of TCM-VAE, we compare VAE decoder variants: the vanilla image decoder (Esser et al., 2020), the 3D convolution decoder, and the temporally fine-tuned decoder (Blattmann et al., 2023). Models are trained with a subset of 100,000 videos from the HD-100M (Xue et al., 2021) dataset to avoid watermarks. We assess video reconstruction using PSNR and SSIM in the YUV color space on the UCF-101 test dataset comprising 3,783 videos. We evaluate text-to-video generation with VersVideo-L using FVD, CLIPSIM, and FC.

In Table 3, the vanilla image decoder overlooks temporal inter-frame dependencies, leading to only moderate reconstruction performance. Previous studies have demonstrated that incorporating 3D convolution or adding temporal layers to the standard image decoder can significantly improve video reconstruction and generation. Our designs of the proposed TCM-VAE collectively contribute to a superior overall performance. Specifically, our TCM-VAE achieves an improvement in video reconstruction, with an increase of 0.643 dB in PSNR and 0.0124 in SSIM. Furthermore, the text-to-video generation achieves an improvement of 16.95 in IS, an improvement of 0.015 in FC, and a reduction of 155 in FVD. These results highlight the effectiveness of our modifications in enhancing both the fidelity of video reconstruction and the quality of text-to-video generation.

## 4 CONCLUSION

In response to the oversimplified spatial-temporal modeling prevalent in most video diffusion models, we introduce a temporal-enhanced diffusion model that incorporates multi-excitation convolution and MoE attention, without significantly increasing the computation costs. Additionally, we augment the VAE decoder to reconstruct the latent space into pixel space and employ a unified control model to accommodate diverse conditions.

**Limitation and future work**. (i) The generated videos sometimes contain watermarks due to the training dataset WebVid-10M. To ensure high-quality generation, it is necessary to have watermark-free videos with textual descriptions. (ii) Our primary focus is on generating videos of a fixed length (e.g., $L = 24$). However, an interesting future direction would be to extend our method for long video synthesis, considering clip-by-clip generation. (iii) VersVideo inherits certain common challenges from image diffusion, including hand generation and text alignment. Solutions employed in image generation could serve as inspiration for video generation. (iv) The stability of image-to-video generation is not perfect. The input image cannot be consistently maintained throughout the videos. It requires additional control to preserve the spatial appearance.

## REPRODUCTIVITY STATEMENT

We provide details (e.g., hyperparameter, model, and optimizer) and experiment setups (e.g., datasets, metrics) in Section 3.1 and Appendix A, B, and C.

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

Table 4: `VersVideo` network details.

| | VersVideo-L | VersVideo-H |
|---|---|---|
| **Architecture** | LDM | LDM |
| video resolution | $16 \times 256 \times 256 \times 3$ | $24 \times 576 \times 320 \times 3$ |
| VAE factor $f$ | 8 | 8 |
| $z$-shape | $16 \times 32 \times 32 \times 4$ | $24 \times 72 \times 40 \times 4$ |
| channels | 128 | 320 |
| Depth | 2 | 2 |
| Channel multiplier | $1, 2, 4, 4$ | $1, 2, 4, 4$ |
| Attention resolutions | $32, 16, 8$ | $72, 36, 18$ |
| Number of heads | 8 | 8 |
| **Cross-Attention** | | |
| Embedding dimension | 768 | 768 |
| CA resolutions | $32, 16, 8$ | $72, 36, 18$ |
| CA sequence length | 77 | 77 |
| **MEC kernel size** | | |
| 2D spatial + 1D temporal | (3,3) + (3,1,1) | (3,3) + (3,1,1) |
| Temporal shift | (3,) | (3,) |
| Spatial-temporal excitation | (3,3,3) | (3,3,3) |
| Channel excitation squeeze | (1, 1) | (1, 1) |
| Channel excitation temporal | (3,) | (3,) |
| Channel excitation expand | (1, 1) | (1, 1) |
| Motion excitation squeeze | (1, 1) | (1, 1) |
| Motion excitation temporal | (3, 3) | (3, 3) |
| Motion excitation expand | (1, 1) | (1, 1) |
| **MoE-STA** | | |
| Number of Spatial FFNs | 32 | 32 |
| Number of Temporal FFNs | 4 | 4 |
| **Training** | | |
| Parameterization | **v** | **v** |
| # train steps | 100 k | 100 k |
| Learning rate | $5 \times 10^{-5}$ | $5 \times 10^{-5}$ |
| Batch size per GPU | 16 | 2 |
| # GPUs | 16 | 32 |
| GPU-type | $A100 - 40$ GB | $A100 - 40$ GB |
| $p_{\text{drop}}$ | 0.1 | 0.1 |
| **Diffusion Setups** | | |
| Diffusion steps | 1000 | 1000 |
| Noise schedule | Linear | Linear |
| $\beta_0$ | 0.00085 | 0.00085 |
| $\beta_T$ | 0.0120 | 0.0120 |
| **Sampling** | | |
| Sampler | DDIM | DDIM |
| Steps | 30 | 30 |
| CFG scale | 7.5 | 7.5 |
| **Total Parameters during Inference** | 500 M | 2 B |

## A   VERSVIDEO NETWORK DETAILS

The paper introduces two distinct variants of video diffusion models, referred to as `VersVideo-L` and `VersVideo-H`. The primary distinction between these two models lies in their dimensions: `VersVideo-L` has a dimension of 128, while `VersVideo-H` has a dimension of 320, as detailed in Table 4. In the ResBlock of the LDM, we have embedded Multi-Excitation Convolutions (MECs), which replace the original 2D convolution designs used for image diffusion. These MECs operate in parallel, providing a more efficient approach. Other parameters such as upsampling or downsampling, as well as input or output channels, remain consistent with the original design. Regarding the MoE Spatial-Temporal Attention (MoE-STA), we have established 32 Feed-Forward Networks (FFNs), or 'experts', for spatial attention, and an additional 4 FFNs for temporal attention during the training phase. It's important to note that during the inference stage, all experts are averaged to produce a single FFN.

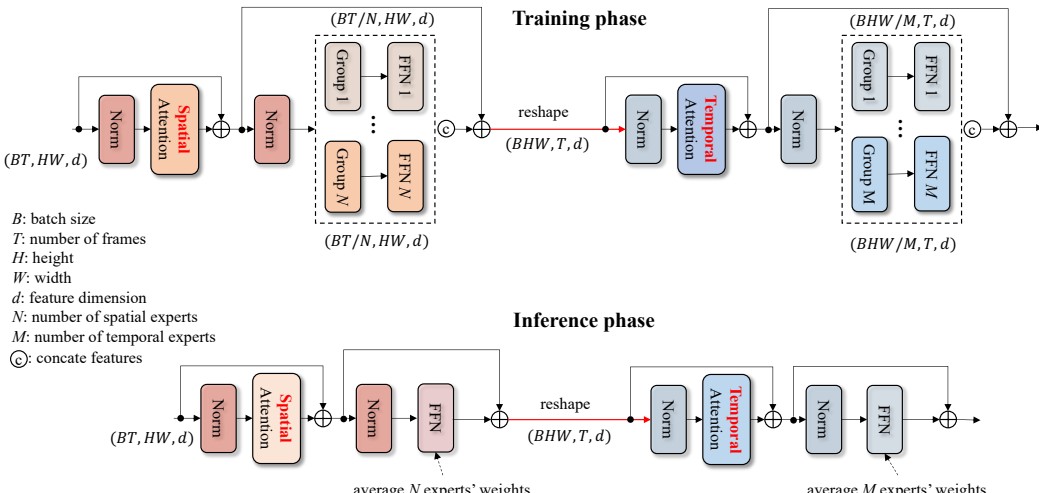

Figure 7: The proposed MoE for the spatial-temporal transformer. In both spatial and temporal transformers, we employ multiple FFN experts during the training phase. These experts are trained using data that is randomly allocated to them. After each training iteration, the weights of the experts are shared amongst each other at an exchange rate denoted by $\lambda$ in Eq. (7). After training, all the experts are averaged to ensemble into a FFN, without increasing the inference costs.

# B MOE-STA

Section 2.2 outlines the proposed MoE for spatial-temporal transformers. In this section, we provide additional details to enhance understanding.

The role of the FFN in transformers is to process the information aggregated by the attention mechanism. It can learn to recognize and generate more intricate patterns based on the information it receives from the attention block. The most recent prevalent training approach for transformers involves replacing the FFN layer with a sparse MoE (Lepikhin et al., 2020; Du et al., 2021; Fedus et al., 2021). The sparse MoE includes multiple expert FFNs, each with unique weights, and a trainable routing network, as shown in Fig. 8. During both the training and inference phases, this routing network selects a sparse set of experts for each input, enabling efficient scalability of transformer models through sparse computation.

As shown in Fig. 7, we introduce a new MoE scheme that trains experts without a router network, reducing the number of parameters introduced by the router and simplifying training. Our MoE assigns tokens to experts by randomly partitioning them into groups. At the end of each training iteration, we perform weight averaging on each MoE's experts. After training, we can average the experts of each MoE into a single FFN, without increasing inference cost.

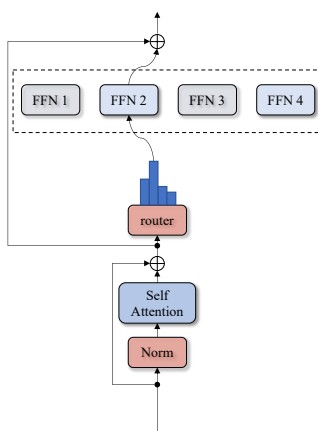

Figure 8: Sparse MoE.

Why does this MoE work? Merging weights of models for better performance is widely acknowledged in large language models (Jin et al., 2022) and image classification (Wortsman et al., 2022; Ainsworth et al., 2022). This simple yet effective idea has been proven to improve accuracy and robustness. Similarly, our proposed MoE merges experts' weights to efficiently enhance the performance of the spatial-temporal transformer. Unlike conventional model ensembles, we average the model weights without incurring any additional inference or memory costs.

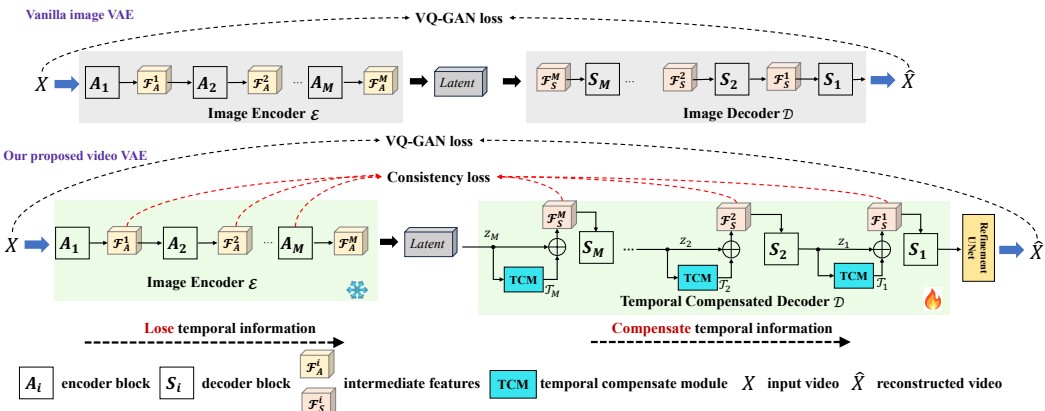

Figure 9: The comparison between the image VAE and the proposed TCM-VAE. Unlike the image VAE, which encodes and reconstructs image frames individually along the temporal dimension, we introduce TCMs and a refinement UNet to enhance inter-frame consistency and video quality.

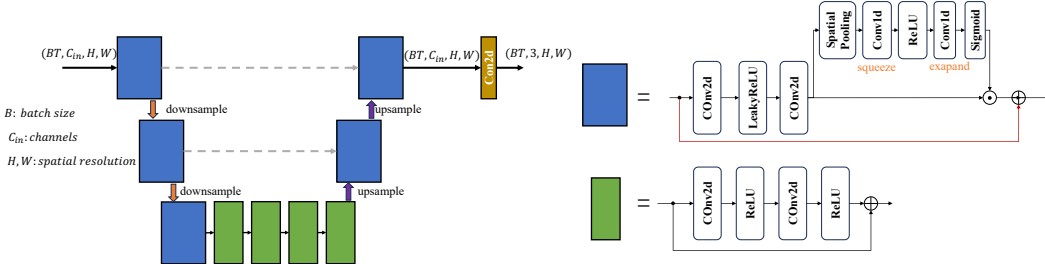

Figure 10: Refinement UNet of TCM-VAE.

## C    TCM-VAE

### C.1    NETWORK DETAILS

There are two critical components in TCM-VAE: the Temporal-Compensated Module (TCM) and the refinement UNet. We choose VQ-GAN (Esser et al., 2020) as the image VAE backbone for TCM-VAE and train the decoder along with TCMs and a refinement UNet.

The encoding process of TCM is identical to the image encoding. Given an input video $X$ of shape $(B, C, T, H, W)$, we reshape the time axis to the batch axis $(BT, C, H, W)$, and then process the video as a batch of images:

$$z = \mathcal{E}(X) = A_M \left( \cdots \left( A_2 \left( A_1 (X) \right) \right) \right). \tag{11}$$

The decoder $\mathcal{D}$ with TCMs embedded can be represented as:

$$\hat{X} = \mathcal{D}(z) = R \left( S_1 \left( \cdots \left( S_{M-1} \left( S_M (\mathcal{F}_S^M) \right) \right) \right) \right), \tag{12}$$

where $R$ is the refinement UNet. $\mathcal{F}_S^M$ is the temporal compensated features using TCMs.

We denote $\mathcal{T}_i$ as the $i$-th TCM block which consists of sequences of convolution layers and activations. We implement all TCMs with the MEC block as detailed in Section 2.2 because it is computationally efficient and also has strong temporal ability. The intermediate features are:

$$\mathcal{F}_S^{M-1} = S_M \left( \mathcal{T}_M (z_M) + z_M \right)$$
$$\cdots$$
$$\mathcal{F}_S^1 = S_1 \left( \mathcal{T}_1 (z_1) + z_1 \right), \tag{13}$$

given the input of decoder $z_M = z$.

It's important to note that in TCM-VAE, all tensors outside the TCM are 4D, specifically $(N \times T, C, H, W)$. Initially, we transform the input 4D tensor into a 5D tensor $(N, T, C, H, W)$ before it's introduced to the TCM, which allows for operations on a specific dimension within the TCM. After this, the 5D output tensor is converted back to a 4D tensor before it's passed to the subsequent TCM-VAE decoder blocks.

The refinement UNet is constructed using residual blocks, and notably, does not include any attention blocks. A comprehensive depiction of the network can be found in Figure 10.

### C.2 VISUAL QUALITY

The VAE from pre-trained image models is often reused for most existing video latent diffusion models. However, the image VAE is a significant contributor to video flickering and inconsistency. To illustrate this issue, we present several visual examples. We compare the widely-used image VAE for StableDiffusion with our proposed TCM-VAE.

As for image VAE, we encode the video frames independently and then reconstruct them back into a video using an image decoder. For TCM-VAE, the encoding process remains unchanged, but we employ the TCM-VAE decoder with temporal awareness.

### C.3 PLUG-AND-PLAY FOR OTHER VIDEO GENERATION MODELS

TCM-VAE offers another advantage in terms of compatibility. Since we freeze the image encoder during training, it is seamless to integrate TCM-VAE with various video diffusion models. This plug-and-play solution allows for easy integration with most existing models. For example, we apply TCM-VAE to VideoComposer (Wang et al., 2023d) and Videofusion (Luo et al., 2023), and the flicker artifacts can be largely alleviated.

## D ABLATION STUDY OF MEC

MEC is a novel approach for spatial-temporal convolutions with dimension pooling across different axes, which avoids the over-simplification of standard 3D operations that often limits spatial-temporal performance in existing video diffusion models. It involves using multi-excitation paths, which include factorized 2D spatial + 1D temporal convolution (2D+1D), spatial-temporal excitation (STE), channel excitation (CE), and motion excitation (ME).

In Section 2.2, we provide a detailed description of MEC. Additionally, an ablation study is conducted in Table 5 to validate each excitation path of MEC, which includes the MoE-STA component. Various versions of the `VersVideo-L` model are trained with and without these paths, followed by an evaluation of their class-conditional generation using the UCF-101 dataset. The results suggest that the excitation modules improve the performance of MEC. When integrating all four excitation paths, the MEC demonstrates superior performance in both IS and FVD.

## E USER STUDY

To conduct the user study, we selected four widely acknowledged and open-sourced models, namely Text2video-zero (Khachatryan et al., 2023), VideoFusion (Luo et al., 2023), Zeroscope (Hugging Face, 2023), and VideoComposer (Wang et al., 2023d), for visual comparison. The evaluation set consisted of 30 video prompts, and 16 evaluators compared the inter-frame consistency and overall

Table 5: Ablation of MEC designs on UCF101.

| 2D+1D | STE | CE | ME | IS ↑ | FVD ↓ |
|-------|-----|-----|-----|------|-------|
|  | ✓ | ✓ | ✓ | 69.5 | 249 |
| ✓ |  | ✓ | ✓ | 68.0 | 264 |
| ✓ | ✓ |  | ✓ | 67.7 | 266 |
| ✓ | ✓ | ✓ |  | 68.3 | 260 |
| ✓ | ✓ | ✓ | ✓ | 72.9 | 207 |

Figure 11: **VersVideo v.s. prior work** in text-to-video in terms of inter-frame consistency and overall quality win-rates evaluated by majority score of human evaluator preferences.

video quality (including inter-frame consistency, alignment of text-video, and perceptual factors) between two videos - one from competing methods and one from our method, shown in a random sequence. Figure 11 shows that our method, VersVideo, was preferred over VideoFusion $83.3\%$ of the time in terms of video quality, and over VideoComposer $88.5\%$ of the time. Our method also performed significantly better than the baseline methods in terms of inter-frame consistency in the user study.

