# OpenReview forum: "VersVideo: Leveraging Enhanced Temporal Diffusion Models for Versatile Video Generation"
_ICLR.cc/2024/Conference — ICLR 2024 poster_

### Official Review · Reviewer_Cov9 · 2023-10-29

**Soundness:** 3 good
**Presentation:** 3 good
**Contribution:** 2 fair
**Rating:** 6
**Confidence:** 3

**Summary:**

This work introduces a diffusion-based method for video generation. The paper proposes the VersVideo model that leverages multi-excitation spatial-temporal convolution, a Mixture of Experts attention blocks, and a temporal compensated decoder to address the challenges associated with spatial-temporal dynamics. Additionally, it integrates the ControlNet model for versatile video generation, allowing for a wider range of visual conditions.

**Strengths:**

1. This paper is well-motivated and the proposal method is technically sound.
2. The proposed method combines several existing powerful techniques to learn better spatiotemporal features.
3. The proposed method achieves comparable performance on the MSR-VTT dataset.

**Weaknesses:**

1. The proposed method directly combines existing techniques, such as excitation network, MoE to enhance spatiotemporal feature representation, which makes the novelty a bit weak. The proposed Multi-Excitation Convolution applies excitation convolution multiple times. Original excitation convolution should be cited.

2. From Table 1, the proposed fails to achieve better performance than Make-A-Video. Even compared with the methods trained on WebVid-10M, the performance improvement is marginal.

3. From Table 2, the performance of the proposed method is a lot worse than the previous STOA method, e.g., VideoFusion. The author explains this by "the VersVideo-L model has a significantly smaller parameter of 500M". Why not use a larger model to verify the performance? Moreover, 1) Make-A-Video also reports performance on the UCF dataset, the results should be included; 2) the author should also provide performance under the same resolution as previous methods for better comparison.

4. Results in Table 2 and Table 3 are from different training datasets. Better to keep it consistent for better comparison.

**Questions:**

see weakness part.

---

> ### Author Response · Authors · 2023-11-20
> **Response to Reviewer Cov9 [1/4]**
>
> The reviewer **Cov9** primarily focuses on the significance of our contributions and the performance of our model in experiments. In response, we provide a clear explanation of our contributions, acknowledge the related works that inspired our methodology, and thoroughly analyze the performance of our model in comparison to other competing methods used in the experiment.

---

> ### Author Response · Authors · 2023-11-20
> **Response to Reviewer Cov9 [2/4]**
>
> >Q1: The proposed method directly combines existing techniques, such as excitation network, MoE to enhance spatiotemporal feature representation, which makes the novelty a bit weak. The proposed Multi-Excitation Convolution applies excitation convolution multiple times. Original excitation convolution should be cited.
>
> Thank you for your expertise and the time you dedicated to reviewing our paper.
>
>
> In this paper, we tackle three major challenges related to diffusion models for video generation.
> - Firstly, we focus on the denoising UNet, which is the core component responsible for modeling spatial-temporal dynamics.
> - Secondly, we address the VAE's limitations, which lead to flicker artifacts and inter-frame inconsistency caused by information loss during video encoding into latents and reconstruction into pixel space.
> - Finally, we aim to achieve controllable generation, allowing for video generation under a range of conditions.
>
>
> Our contributions strive to address these challenges:
> - We enhance the spatial-temporal capability of the denoising UNet with Multi-Excitation Convolution (MEC) and Mixture of Expert (MoE) spatial-temporal transformers, as shown in Table 1 of the paper.
> - We build the TCM-VAE for video decoding to reduce information loss and increase inter-frame consistency, as demonstrated in Table 3 and visually showcased in [video generation demo](https://docs.google.com/document/d/e/2PACX-1vQyICnc6WH8CgCCRuX0avsOI7MoO4gyPRQEro-v1UGu0nI60CTCiDksLUXOV5y_A597X41rL5mQdyER/pub) and [video reconstruction demo](https://anonymous-pages.github.io/video_demos/).
> - We propose a unified controllable video generation strategy for text, visual, image, and style conditions, as demonstrated in Fig. 6 and visually showcased in the [demo](https://anonymous-pages.github.io/video_demos/).
>
>
> **Multi-Excitatin Convolution (MEC) motivation.** In contrast to existing video diffusion models, which often restrict spatial-temporal performance due to the over-simplification of standard 3D operations, we propose multi-excitation paths for spatial-temporal convolutions with dimension pooling across different axes.
>
> The proposed MEC is constructed by assembling four paths in parallel, allowing for the **activation of multi-type information in videos**. Our design is fundamentally inspired by the squeeze-and-excitation (SE) block [ref-1; ref-2], as it explicitly models channel/temporal interdependencies. Drawing inspiration from these two previous works, we design the excitation paths:
> 1. Factorized 2D spatial + 1D temporal convolution. This path is a **simplified adaptation of 3D convolution** and is the only temporal module widely used in most existing video diffusion models [ref-3; ref-4; ref-5].
> 2. Spatial-temporal excitation. This path pools the 5-D tensor $(B, C, T, H, W)$ along channel dimension, resulting in the **spatial-temporal activation** mask $(B, 1, T, H, W)$ which is later processed with 3D convolution.
> 3. Channel excitation. This path get global spatial information $(B, C, T, 1, 1)$ of the input feature $(B, C, T, H, W)$ by spatial average pooling.
> 4. Motion excitation. The **motion information** is modeled by adjacent frames using the same squeeze and unsqueeze strategy. Related works have explored this, aiming to model motion information based on the feature level instead of the pixel level [ref-2; ref-6].
>
>
>
> *Reference*:
>
> [ref-1] Hu, J., Shen, L. and Sun, G., 2018. Squeeze-and-excitation networks. In Proceedings of the IEEE conference on computer vision and pattern recognition (pp. 7132-7141).
>
> [ref-2] Li, Y., Ji, B., Shi, X., Zhang, J., Kang, B. and Wang, L., 2020. Tea: Temporal excitation and aggregation for action recognition. In Proceedings of the IEEE/CVF conference on computer vision and pattern recognition (pp. 909-918).
>
> [ref-3] Wang, X., Yuan, H., Zhang, S., Chen, D., Wang, J., Zhang, Y., Shen, Y., Zhao, D. and Zhou, J., 2023. VideoComposer: Compositional Video Synthesis with Motion Controllability. arXiv preprint arXiv:2306.02018.
>
> [ref-4] Blattmann, A., Rombach, R., Ling, H., Dockhorn, T., Kim, S.W., Fidler, S. and Kreis, K., 2023. Align your latents: High-resolution video synthesis with latent diffusion models. In Proceedings of the IEEE/CVF Conference on Computer Vision and Pattern Recognition (pp. 22563-22575).
>
> [ref-5] Guo, Y., Yang, C., Rao, A., Wang, Y., Qiao, Y., Lin, D. and Dai, B., 2023. Animatediff: Animate your personalized text-to-image diffusion models without specific tuning. arXiv preprint arXiv:2307.04725.
>
> [ref-6] Jiang, B., Wang, M., Gan, W., Wu, W. and Yan, J., 2019. Stm: Spatiotemporal and motion encoding for action recognition. In Proceedings of the IEEE/CVF International Conference on Computer Vision (pp. 2000-2009).

---

> ### Author Response · Authors · 2023-11-20
> **Response to Reviewer Cov9 [3/4]**
>
> >Q2: From Table 1, the proposed fails to achieve better performance than Make-A-Video. Even compared with the methods trained on WebVid-10M, the performance improvement is marginal.
>
> We appreciate your thorough review and the effort you put into providing detailed feedback.
>
> In Table 1, we employ the Frechet Video Distance (FVD) and CLIP similarity (CLIPSIM) as evaluation metrics for text-to-video generation. These two metrics are commonly utilized in most of video generation studies. The FVD metric assesses the similarity between real and generated videos, taking into account both video quality and motion dynamics. On the other hand, CLIPSIM measures the video quality and text alignment by averaging the CLIP similarity score of all generated frames.
>
>
> Our proposed *VersVideo-H* diffusion model, trained with WebVid-10M, consists of 2B parameters. In contrast, **Make-A-Video serves as a strong competing method with significantly larger parameters and training data.** It encompasses 9.72B parameters in total, including a 3.1B text-to-video diffusion model, a 1.3B prior model, a 3.1B frame interpolation model, and more. Make-A-Video is trained using 2.3B images, WebVid-10M videos, and a 10M video subset from HD-VILA-100M. As a result, Make-A-Video achieves the highest CLIPSIM score of $0.3049$, while our proposed VersVideo-H model achieves a slightly lower score of $0.3014$.
>
> Moreover, it's worth noting that Make-A-Video did not provide results for the FVD metric for motion dynamic evaluation.
>
>
> When comparing our proposed VersVideo model with the most recent models trained using WebVid-10M, it outperforms them all. For instance, Videocompoer (1.85B) and Videofusion (1.83B) have comparable model sizes, but their performance falls short with FVD scores of $580$ and $581$, and CLIPSIM scores of $0.2932$ and $0.2795$, respectively. In contrast, our proposed model achieves superior results with an FVD score of $421$ and a CLIPSIM score of $0.3014$. The second-best model that comes close to ours is SimDA that is an up-to-date concurrent work within two months of our submission. SimDA achieves an FVD score of $456$ and a CLIPSIM score of $0.2945$. These results highlight the performance of our VersVideo model over other recent video generation models.
>
>
>
> >Q3: From Table 2, the performance of the proposed method is a lot worse than the previous STOA method, e.g., VideoFusion. The author explains this by "the VersVideo-L model has a significantly smaller parameter of 500M". Why not use a larger model to verify the performance?
>
> Thank you for your thorough review.
>
>
> In this paper, we introduce two variants of *VersVideo*: *VersVideo-L* (500M) and *VersVideo-H* (2B). The primary purpose of *VersVideo-L* is for fast prototyping and conducting ablation studies.  *VersVideo-H* is trained for text-to-video generation and controllable generation tasks. If we were to use *VersVideo-H* for prototyping and ablation studies, it would require training several models with billions of parameters from scratch. This would result in an exceptionally high cost for validating model design and optimizing the network. Therefore, we choose to conduct the ablation study using *VersVideo-L* to validate our model design, and subsequently scale up the model to 2B parameters.

---

> > ### Author Response · Authors · 2023-11-20
> > **Response to Reviewer Cov9 [4/4]**
> >
> > >Q4: Moreover, 1) Make-A-Video also reports performance on the UCF dataset, the results should be included; 2) the author should also provide performance under the same resolution as previous methods for better comparison.
> >
> > Thank you for your suggestions.
> >
> >
> >
> > We update Table 2 in the paper and include the performance of Make-a-video. Both the MEC and MoE-STA have demonstrated their effectiveness as design elements in improving the performance of the video diffusion model. These elements consistently contribute to performance enhancements. Although the *VersVideo-L* model does not surpass state-of-the-art models like Make-a-Video and VideoFusion, which boast billions of parameters, it is worth noting that the *VersVideo-L* model has a significantly smaller parameter count of 500 M. Notably, the *VersVideo-H* model achieves an IS score of $81.3$ and an FVD score of $119$, making it the second-best performing model after Make-A-Video.
> >
> > | Method | Resolution | $\mathrm{IS} \uparrow$ | $\mathrm{FVD} \downarrow$ |
> > | :---: | :---: | :---: | :---: |
> > | TGAN (Saito et al, 2016) | $64 \times 64$ | 15.83 | - |
> > | MoCoGAN-HD (Tulyakov et al., 2017) | $128 \times 128$ | 12.42 |  |
> > | CogVideo (Hong et al., 2022) | $160 \times 160$ | 50.46 | 626 |
> > | DVD-GAN (Clark et al., 2019) | $128 \times 128$ | 32.97 | - |
> > | TATS (Ge et al., 2022) | $128 \times 128$ | 79.28 | 332 |
> > | VideoFusion (Luo et al., 2023) | $128 \times 128$ | 80.03 | 173 |
> > | **Make-A-Video (Singer et al, 2022)** | $256 \times 256$ | **82.55** | **81.25** |
> > | VersVideo-L (baseline) | $256 \times 256$ | 60.2 | 355 |
> > | VersVideo-L (+ MEC) | $256 \times 256$ | 68.8 | 287 |
> > | VersVideo-L (+ MoE-STA) | $256 \times 256$ | 63.5 | 302 |
> > | VersVideo-L (+ MEC + MoE) | $256 \times 256$ | 72.9 | 207 |
> > | **VersVideo-H** | $576 \times 320$ | **81.3** | **119** |
> >
> >
> > In terms of resolution, it should be noted that FVD features are extracted with an I3D model trained on Kinetics-400 and the Inception Score (IS) calculated with a C3D model [ref-1]. During metric computation, the C3D model resizes all videos to a spatial resolution of $112\times112$, as determined by its internal structure.  Consequently, we believe that there would be no substantial difference in the feature level between resolutions of $256\times256$ and $128\times128$. If you would like to explore additional evaluation issues concerning FVD and IS, please refer to the Section C in the Appendix of StyleGAN-V [ref-2].
> >
> >
> > *Reference*:
> > [ref-1] Tran, D., Bourdev, L., Fergus, R., Torresani, L. and Paluri, M., 2015. Learning spatiotemporal features with 3d convolutional networks. In Proceedings of the IEEE international conference on computer vision (pp. 4489-4497).
> > [ref-2] Skorokhodov, I., Tulyakov, S. and Elhoseiny, M., 2022. Stylegan-v: A continuous video generator with the price, image quality and perks of stylegan2. In Proceedings of the IEEE/CVF Conference on Computer Vision and Pattern Recognition (pp. 3626-3636).
> >
> > >Q5: Results in Table 2 and Table 3 are from different training datasets. Better to keep it consistent for better comparison.
> >
> > Thank you for your thoughtful comments. Table 2 and Table 3 represent two distinct experiments that utilize different training datasets for specific reasons.
> >
> > Table 2 aims to evaluate the effectiveness of the **denoising UNet** design, which incorporates multi-excitation convolution (MEC) and Mixture of Experts (MoE) attention. To conduct this experiment, we utilize **text-video pairs** sourced from WebVid-10M. This dataset offers high-quality textual descriptions for videos and encompasses a diverse range of content.
> >
> > On the other hand, Table 3 aims to train the **TCM-VAE** for video reconstruction using only video data.   Unlike Table 2, which requires text-video pairs, video captions are not needed to train the TCM-VAE. We did not use WebVid-10M because its videos contain watermarks. Therefore, we obtained 100,000 high-quality videos from the HD-100M dataset for this experiment.

---

### Official Review · Reviewer_CWze · 2023-10-31

**Soundness:** 3 good
**Presentation:** 3 good
**Contribution:** 3 good
**Rating:** 6
**Confidence:** 3

**Summary:**

This paper addresses the challenge of creating stable and controllable videos by proposing a versatile video generation model. In contrast to existing video diffusion models that often limit spatial-temporal performance due to oversimplification of standard 3D operations, VersVideo introduces multi-excitation paths for spatial-temporal convolutions with dimension pooling across different axes and multi-expert spatial-temporal attention blocks. This approach significantly improves the model's spatial-temporal performance without increasing training and inference costs. To address information loss during the transformation from pixel space to latent features and back, temporal modules are incorporated into the decoder to maintain inter-frame consistency. The paper also presents a unified ControlNet model suitable for various conditions, such as image, Canny, HED, depth, and style.

**Strengths:**

1. The introduction of multi-excitation paths and multi-expert spatial-temporal attention blocks enhances the model's spatial-temporal capabilities without significantly increasing computational cost. This is particularly impressive given the complexity of 3D operations in video generation. The solution proposed for minimizing the issue of information loss, mainly through the incorporation of the temporal module, helps achieve better inter-frame consistency, which is crucial for maintaining the temporal coherence of generated videos. The development of a unified ControlNet provides a more versatile approach to handle various conditions, leading to broad applicability.
2. The quantitative improvements of the proposed method are notable. The related ablations offer reliable evidence about the effectiveness of the whole system.

**Weaknesses:**

1. Leveraging pre-trained models could limit the model's versatility and the ability to generalize across different datasets. Is it possible to validate the proposed designs to other SD models?
2. The notable quantitative and qualitative improvements are appreciated. However, some common issues are not discussed in the paper, e.g., hand generation, multiple object generation, instruction following, and how to respond to relative location descriptions, etc. If these cases cannot be well-addressed, then they should be included in a limitation discussion.

**Questions:**

1. It might be beneficial to include a section discussing the limitations of your current approach and possible future work to address these limitations.
2. If possible, consider releasing the code or at least providing more detailed implementation notes. This could help others replicate your results and build upon your work.

---

> ### Author Response · Authors · 2023-11-20
> **Response to Reviewer CWze [1/3]**
>
> The reviewer **CWze** primarily focused on the limitations and future scope of our work. The reviewer also encouraged us to add more details about the implementation. In response, we clarified our contributions and discussed the potential application of our work to other related studies. Additionally, we added more implementation details about the proposed MoE spatial-temporal transformer and TCM-VAE to ensure reproducibility.
>
> >Q1: Leveraging pre-trained models could limit the model's versatility and the ability to generalize across different datasets. Is it possible to validate the proposed designs to other SD models?
>
>
> Thank you for your review and valuable suggestions.
>
> Incorporating temporal structures for video generation is a widely adopted approach to enhance existing text-to-image models, such as Stable Diffusion (SD). Several methods have been employed in this regard, including training the entire network [ref-1; ref-2; ref-3] or freezing the pretrained image network to ensure compatibility with other SD models [ref-4; ref-5; ref-6].
>
>
> Our designs can be utilized for other video diffusion models that are based on SD in three different ways:
> 1. The over-simplified temporal resblock/transformer can be replaced with the proposed MEC and MoE transformer of the denoising UNet.
> 2.  The image VAE can be replaced by our proposed TCM-VAE. For example, we can enhance the image VAE decoder of *VideoFusion* and *VideoComposer* by using our TCM-VAE decoder, without additional training. This module significantly reduces flicker artifacts, as shown in our [demo](https://docs.google.com/document/d/e/2PACX-1vQyICnc6WH8CgCCRuX0avsOI7MoO4gyPRQEro-v1UGu0nI60CTCiDksLUXOV5y_A597X41rL5mQdyER/pub).
> 3.  Designing an unified control strategy outlined in our paper, rather than developing individual motion structure conditions with multiple individual ControlNets for video generation.
>
>
> Reference:
>
> [ref-1] Esser, P., Chiu, J., Atighehchian, P., Granskog, J. and Germanidis, A., 2023. Structure and content-guided video synthesis with diffusion models. In Proceedings of the IEEE/CVF International Conference on Computer Vision (pp. 7346-7356).
>
> [ref-2] Wu, J.Z., Ge, Y., Wang, X., Lei, S.W., Gu, Y., Shi, Y., Hsu, W., Shan, Y., Qie, X. and Shou, M.Z., 2023. Tune-a-video: One-shot tuning of image diffusion models for text-to-video generation. In Proceedings of the IEEE/CVF International Conference on Computer Vision (pp. 7623-7633).
>
> [ref-3] Zhou, D., Wang, W., Yan, H., Lv, W., Zhu, Y. and Feng, J., 2022. Magicvideo: Efficient video generation with latent diffusion models. arXiv preprint arXiv:2211.11018.
>
> [ref-4] Khachatryan, L., Movsisyan, A., Tadevosyan, V., Henschel, R., Wang, Z., Navasardyan, S. and Shi, H., 2023. Text2video-zero: Text-to-image diffusion models are zero-shot video generators. arXiv preprint arXiv:2303.13439.
>
> [ref-5] Blattmann, A., Rombach, R., Ling, H., Dockhorn, T., Kim, S.W., Fidler, S. and Kreis, K., 2023. Align your latents: High-resolution video synthesis with latent diffusion models. In Proceedings of the IEEE/CVF Conference on Computer Vision and Pattern Recognition (pp. 22563-22575).
>
> [ref-6] Guo, Y., Yang, C., Rao, A., Wang, Y., Qiao, Y., Lin, D. and Dai, B., 2023. Animatediff: Animate your personalized text-to-image diffusion models without specific tuning. arXiv preprint arXiv:2307.04725.

---

> ### Author Response · Authors · 2023-11-20
> **Response to Reviewer CWze [2/3]**
>
> >Q2: The notable quantitative and qualitative improvements are appreciated. However, some common issues are not discussed in the paper, e.g., hand generation, multiple object generation, instruction following, and how to respond to relative location descriptions, etc. If these cases cannot be well-addressed, then they should be included in a limitation discussion.
>
> Thank you for your insightful and detailed review of this point.
>
> The common issues you mentioned are widely acknowledged in image diffusion generation and are also inherent in most video diffusion models. For instance, it is well-known that diffusion models struggle to create realistic hands, mainly due to the inherent complexity of the human hand and the variability and diversity of the hands in the training dataset.
>
> Another challenge is text alignment, which involves multi-object generation and relative location assignment. Recently, significant progress has been made in this direction by Dalle-3 [ref-1] and other other works [ref-2] on image generation. They have improved text-to-image models by training them on enhanced captions auto-generated by a robust image-to-text model. This approach largely alleviates the text alignment problem. Similarly, addressing these specific problems in video generation necessitates specially tailored solutions. For instance, utilizing more precise textual descriptions for training videos or incorporating a wider range of hand data.
>
> Since *VersVideo* utilizes the WebVid-10M dataset, which contains video-text data from the internet, additional efforts in data curation/collection and related techniques for image generation are necessary to address these challenges. We acknowledge this limitation in our paper.
>
>
> Reference:
>
> [ref-1] Betker, James, Gabriel Goh, Li Jing, † TimBrooks, Jianfeng Wang, Linjie Li, † LongOuyang, † JuntangZhuang, † JoyceLee, † YufeiGuo, † WesamManassra, † PrafullaDhariwal, † CaseyChu, † YunxinJiao and Aditya Ramesh, 2023. Improving Image Generation with Better Captions.
>
> [ref-2] Segalis, E., Valevski, D., Lumen, D., Matias, Y. and Leviathan, Y., 2023. A Picture is Worth a Thousand Words: Principled Recaptioning Improves Image Generation. arXiv preprint arXiv:2310.16656.
>
>
> >Q3: It might be beneficial to include a section discussing the limitations of your current approach and possible future work to address these limitations.
>
> Thank you for your insightful comments and suggestions.
>
> Despite its strong generative capabilities and versatile guidance inputs, the proposed *VersVideo* has some limitations in its current state:
> 1. The generated videos sometimes contain watermarks due to the training dataset WebVid-10M. To ensure high-quality generation, it is necessary to have watermark-free videos with textual descriptions.
> 2. Our primary focus is on generating videos of a fixed length (e.g., $L = 24$). However, an interesting future direction would be to extend our method for long video synthesis, considering clip-by-clip generation.
> 3. *VersVideo* inherits certain common challenges from image diffusion, including hand generation and text alignment. Specific solutions employed in image generation could serve as inspiration for video generation.
> 4. The stability of image-to-video generation is not perfect. The input image cannot be consistently maintained throughout the generated video. It requires additional control mechanisms to preserve the spatial appearance of the input image more accurately.
>
> [A snapshot of our revision.](https://drive.google.com/file/d/1f1UdXPXBKIKyhMWgVA8SiF3w7FOx4Nt1/view?usp=sharing)

---

> ### Author Response · Authors · 2023-11-20
> **Response to Reviewer CWze [3/3]**
>
> >Q4: If possible, consider releasing the code or at least providing more detailed implementation notes. This could help others replicate your results and build upon your work.
>
> Thank you for taking the time to review our paper.
>
> We provide details (e.g., hyperparameter, model, and optimizer) and experiment setups (e.g., datasets, metrics) in Section 3.2 and Appendix A, B, and C. We incorporated additional details regarding the key modules to facilitate implementation and enhance understanding.
> 1. MoE spatial-temporal transformer.  In our paper, we introduce a new MoE scheme that trains experts without a router network, reducing the number of parameters introduced by the router and simplifying training. Our MoE assigns tokens to experts by randomly partitioning them into groups. At the end of each training iteration, we perform weight averaging on each MoE's experts. After training, we can simply average the experts of each MoE into a single FFN, without increasing inference cost.  Merging weights of models for better performance is widely acknowledged in large language models and image classification. This simple yet effective idea has been proven to improve accuracy and robustness. Similarly, our proposed MoE merges experts' weights to efficiently enhance the performance of the spatial-temporal transformer. Unlike conventional model ensembles, we average the model weights without incurring any additional inference or memory costs. [A snapshot of our revision](https://drive.google.com/file/d/1WXwEmOOBMi4Mr7dxF2U65-viJ1lf2A5J/view?usp=sharing).
> 2. TCM-VAE. We introduce Temporal Compensation Modules (TCMs) to enhance temporal consistency, and a refinement UNet for quality enhancement, building upon an image pretrained VAE. To optimize training efficiency, we freeze the encoder and concentrate on training the video decoder. We incorporate TCMs into the decoder blocks to infuse temporal information into multi-level intermediate features. The implementation of TCMs leverages the multi-excitation module in Section 2.2.  More details about the TCM-VAE can be found in **Section2.3, page 5** and **Section C of the Appendix**. [A snapshot of our revision.](https://drive.google.com/file/d/1r6jnbaAk_OLFXTVXUyc1ogPlxD7ZxrrB/view?usp=sharing)

---

> ### Author Response · Authors · 2023-11-23
> **Follow-Up Discussion**
>
> Dear Reviewer CWze,
>
> We sincerely value the time and effort you have dedicated to offering your perceptive observations on our paper. After thoroughly examining your feedback, we have formulated responses that we believe will effectively address your concerns.
>
> We have expanded our discussion to further address the limitations of our model, as well as to outline the potential future directions this study could take.
>
> We hope you can reach out to us at your earliest convenience.
>
> Best regards,

---

> > ### Comment · Reviewer_CWze · 2023-11-23
> >
> > Thank you for your diligent efforts in addressing the concerns that were raised. However, after a thorough reconsideration of your paper in light of these responses, my assessment aligns with my initial rating. I encourage you to continue refining and developing your work. Your research is valuable and contributes significantly to the field.

---

### Official Review · Reviewer_eX2J · 2023-11-01

**Soundness:** 2 fair
**Presentation:** 2 fair
**Contribution:** 2 fair
**Rating:** 3
**Confidence:** 2

**Summary:**

The paper proposes VersVideo for video generation. To capture the temporal information, multi-excitation paths for spatial-temporal convolutions and multi-expert spatial-temporal attention blocks. Also, the decoder of VAE is also finetuned to maintain inter-frame consistency. The authors demonstrate the effectiveness of VersVideo in multiple video generation tasks, e.g., text-to-video generation and conditional video generation.

**Strengths:**

1) The paper studies video generation and proposes several blocks to enhance the temporal consistency of generated videos.

2) The generated videos have plausible results.

3) The authors demonstrate the capability of the proposed network in the controllable generation setting.

**Weaknesses:**

1) The details of the MoE attention are not explained very well. In section 2.2, the authors mentioned that they use MoE design to enhance attention. If my understanding is correct, the attention is performed on all channels, but the feedforward network after attention is applied to different groups. If the MoE attention is applied like this, how can the feature or attention be enhanced?

2) Also, several details are missing for the temporal compensate decoder. In the paper, the authors do not mention what S1 stands for.

3) For the Multi-excitation conv, the motivation for using four branches is not addressed clearly. The effectiveness of each branch is also not thoroughly ablated.

**Questions:**

Please see my concerns in the weakness part.

---

> ### Author Response · Authors · 2023-11-20
> **Response to Reviewer eX2J [1/3]**
>
> The reviewer **eX2J** has raised concerns primarily about the methodology details in the paper. To address these concerns, we have added more information about the motivation and implementation of the proposed MoE spatial-temporal transformer and the temporal compensated VAE, with the help of illustrations and visual demonstrations. We have also conducted an ablation study to verify the effectiveness of each path in the multi-excitation convolution.
>
>
>
> > Q1: The details of the MoE attention are not explained very well. In section 2.2, the authors mentioned that they use MoE design to enhance attention. If my understanding is correct, the attention is performed on all channels, but the feedforward network after attention is applied to different groups. If the MoE attention is applied like this, how can the feature or attention be enhanced?
>
> Thank you for highlighting this issue; we apologize for any confusion caused.
>
> Our design might be more appropriate to refer to it as the 'MoE Spatial-Temporal Transformer' rather than 'MoE Spatial-Temporal Attention.'
>
> The role of the FFN in transformers is to process the information aggregated by the attention mechanism. It can learn to recognize and generate more intricate patterns based on the information it receives from the attention block.
>
> The most recent prevalent training approach for transformers involves replacing the FFN layer with a sparse MoE [ref-1; ref-2; ref-3]. The sparse MoE includes multiple expert FFNs, each with unique weights, and a trainable routing network. During both the training and inference phases, this routing network selects a sparse set of experts for each input, enabling efficient scalability of transformer models through sparse computation.
>
> In our paper, we introduce a new MoE scheme that trains experts without a router network, reducing the number of parameters introduced by the router and simplifying training. Our MoE assigns tokens to experts by randomly partitioning them into groups. At the end of each training iteration, we perform weight averaging on each MoE's experts. After training, we can simply average the experts of each MoE into a single FFN, without increasing inference cost.
>
> **Why does this MoE work?** Merging weights of models for better performance is widely acknowledged in large language models [ref-6] and image classification [ref-4; ref-5]. This simple yet effective idea has been proven to improve accuracy and robustness. Similarly, our proposed MoE merges experts' weights to efficiently enhance the performance of the spatial-temporal transformer. Unlike conventional model ensembles, we average the model weights without incurring any additional inference or memory costs.
>
>
>
> Please kindly refer to **Section B in the Appendix** where we have added more details and descriptions of the MoE design. [A snapshot of our revision](https://drive.google.com/file/d/1WXwEmOOBMi4Mr7dxF2U65-viJ1lf2A5J/view?usp=sharing).
>
> _Reference_:
>
> [ref-1] Du, N., Huang, Y., Dai, A.M., Tong, S., Lepikhin, D., Xu, Y., Krikun, M., Zhou, Y., Yu, A.W., Firat, O. and Zoph, B., 2022, June. Glam: Efficient scaling of language models with mixture-of-experts. In International Conference on Machine Learning (pp. 5547-5569). PMLR.
>
> [ref-2] Fedus, W., Zoph, B. and Shazeer, N., 2022. Switch transformers: Scaling to trillion parameter models with simple and efficient sparsity. The Journal of Machine Learning Research, 23(1), pp.5232-5270.
>
> [ref-3] Lepikhin, D., Lee, H., Xu, Y., Chen, D., Firat, O., Huang, Y., Krikun, M., Shazeer, N. and Chen, Z., 2020. Gshard: Scaling giant models with conditional computation and automatic sharding. arXiv preprint arXiv:2006.16668.
>
> [ref-4] Wortsman, M., Ilharco, G., Gadre, S.Y., Roelofs, R., Gontijo-Lopes, R., Morcos, A.S., Namkoong, H., Farhadi, A., Carmon, Y., Kornblith, S. and Schmidt, L., 2022, June. Model soups: averaging weights of multiple fine-tuned models improves accuracy without increasing inference time. In International Conference on Machine Learning (pp. 23965-23998). PMLR.
>
> [ref-5] Ainsworth, S.K., Hayase, J. and Srinivasa, S., Git re-basin: Merging models modulo permutation symmetries, 2022. URL https://arxiv. org/abs/2209.04836.
>
> [ref-6] Jin, X., Ren, X., Preotiuc-Pietro, D. and Cheng, P., 2022. Dataless knowledge fusion by merging weights of language models. arXiv preprint arXiv:2212.09849.

---

> ### Author Response · Authors · 2023-11-20
> **Response to Reviewer eX2J [2/3]**
>
> >Q2: Also, several details are missing for the temporal compensate decoder. In the paper, the authors do not mention what S1 stands for.
>
> Thank you for pointing out this issue. We sincerely apologize for such confusions.
>
> In Fig. 3, $\{A_1, A_2,...,A_M\}$ are encoder blocks; $\{S_1, S_2, ..., S_M\}$ are decoder blocks. See [Fig. 3](https://drive.google.com/file/d/1-OtQhJ5zHNQGo4zTq-BQgAZxIwI9Az1x/view?usp=sharing) in the revised paper.
>
> Most existing video latent diffusion models utilize image VAEs to reconstruct videos from latents. However, an image VAE without temporal modules fails to capture the dependencies between frames. This leads to a loss of temporal information in videos, which in turn results in flickering artifacts and temporal inconsistency.
>
> We introduce Temporal Compensation Modules (TCMs) to enhance temporal consistency, and a refinement UNet for quality enhancement, building upon an image pretrained VAE. To optimize training efficiency, we freeze the encoder and concentrate on training the video decoder. We incorporate TCMs into the decoder blocks to infuse temporal information into multi-level intermediate features. The implementation of TCMs leverages the multi-excitation module in Section 2.2.
>
> More details about the TCM-VAE can be found in **Section2.3, page 5** and **Section C of the Appendix**. [A snapshot of our revision.](https://drive.google.com/file/d/1r6jnbaAk_OLFXTVXUyc1ogPlxD7ZxrrB/view?usp=sharing)
>
> The effectiveness of TCM-VAE in improving video quality is supported by quantitative metrics presented in **Table 3** of the paper, as well as visual examples available in the demo for video generation and reconstruction: [demo for video generation](https://docs.google.com/document/d/e/2PACX-1vQyICnc6WH8CgCCRuX0avsOI7MoO4gyPRQEro-v1UGu0nI60CTCiDksLUXOV5y_A597X41rL5mQdyER/pub) and [demo for video reconstruction](https://anonymous-pages.github.io/video_demos/).

---

> ### Author Response · Authors · 2023-11-20
> **Response to Reviewer eX2J [3/3]**
>
> >Q3: For the Multi-excitation conv, the motivation for using four branches is not addressed clearly. The effectiveness of each branch is also not thoroughly ablated.
>
>
> We appreciate your thorough review and the effort you put into providing detailed feedback.
>
>
> **Multi-Excitatin Convolution (MEC) motivation.** In contrast to existing video diffusion models, which often restrict spatial-temporal performance due to the over-simplification of standard 3D operations, we propose multi-excitation paths for spatial-temporal convolutions with dimension pooling across different axes.
>
> The proposed MEC is constructed by assembling four paths in parallel, allowing for the **activation of multi-type information in videos**. Our design is fundamentally inspired by the squeeze-and-excitation (SE) block [ref-1; ref-2], as it explicitly models channel/temporal interdependencies. Drawing inspiration from these two previous works, we design the excitation paths:
> 1. Factorized 2D spatial + 1D temporal convolution. This path is a **simplified adaptation of 3D convolution** and is the only temporal module widely used in most existing video diffusion models [ref-3; ref-4; ref-5].
> 2. Spatial-temporal excitation. This path pools the 5-D tensor $(B, C, T, H, W)$ along channel dimension, resulting in the **spatial-temporal activation** mask $(B, 1, T, H, W)$ which is later processed with 3D convolution.
> 3. Channel excitation. This path get global spatial information $(B, C, T, 1, 1)$ of the input feature $(B, C, T, H, W)$ by spatial average pooling.
> 4. Motion excitation. The **motion information** is modeled by adjacent frames using the same squeeze and unsqueeze strategy. Related works have explored this, aiming to model motion information based on the feature level instead of the pixel level [ref-2; ref-6].
>
>
>
> **Ablation study of MEC.** An ablation study is conducted to validate each excitation path of MEC, which includes the MoE-STA component. MEC includes factorized 2D spatial + 1D temporal convolution (2D+1D), spatial-temporal excitation (STE), channel excitation (CE), and motion excitation (ME). Various versions of the *VersVideo-L* model are trained with and without these paths, followed by an evaluation of their class-conditional generation using the UCF-101 dataset. The results suggest that the excitation modules improve the performance of MEC. When integrating all four excitation paths, the MEC demonstrates superior performance in both IS and FVD.
>
> | 2D+1D | STE | CE | ME | IS $\uparrow$ | FVD $\downarrow$ |
> | :---: | :---: | :---: | :---: | :---: | :---: |
> |  | $\checkmark$ | $\checkmark$ | $\checkmark$ | 69.5 | 249 |
> | $\checkmark$ |  | $\checkmark$ | $\checkmark$ | 68.0 | 264 |
> | $\checkmark$ | $\checkmark$ |  | $\checkmark$ | 67.7 | 266 |
> | $\checkmark$ | $\checkmark$ | $\checkmark$ |  | 68.3 | 260 |
> | $\checkmark$ | $\checkmark$ | $\checkmark$ | $\checkmark$ | 72.9 | 207 |
>
> Please refer to [**Section D in the Appendix**](https://drive.google.com/file/d/12SAET1FOi6bYzyCwwc3UDzT3qP0OBZHS/view?usp=sharing) for our revision.
>
> *Reference*:
>
> [ref-1] Hu, J., Shen, L. and Sun, G., 2018. Squeeze-and-excitation networks. In Proceedings of the IEEE conference on computer vision and pattern recognition (pp. 7132-7141).
>
> [ref-2] Li, Y., Ji, B., Shi, X., Zhang, J., Kang, B. and Wang, L., 2020. Tea: Temporal excitation and aggregation for action recognition. In Proceedings of the IEEE/CVF conference on computer vision and pattern recognition (pp. 909-918).
>
> [ref-3] Wang, X., Yuan, H., Zhang, S., Chen, D., Wang, J., Zhang, Y., Shen, Y., Zhao, D. and Zhou, J., 2023. VideoComposer: Compositional Video Synthesis with Motion Controllability. arXiv preprint arXiv:2306.02018.
>
> [ref-4] Blattmann, A., Rombach, R., Ling, H., Dockhorn, T., Kim, S.W., Fidler, S. and Kreis, K., 2023. Align your latents: High-resolution video synthesis with latent diffusion models. In Proceedings of the IEEE/CVF Conference on Computer Vision and Pattern Recognition (pp. 22563-22575).
>
> [ref-5] Guo, Y., Yang, C., Rao, A., Wang, Y., Qiao, Y., Lin, D. and Dai, B., 2023. Animatediff: Animate your personalized text-to-image diffusion models without specific tuning. arXiv preprint arXiv:2307.04725.
>
> [ref-6] Jiang, B., Wang, M., Gan, W., Wu, W. and Yan, J., 2019. Stm: Spatiotemporal and motion encoding for action recognition. In Proceedings of the IEEE/CVF International Conference on Computer Vision (pp. 2000-2009).

---

> > ### Author Response · Authors · 2023-11-23
> > **Follow-Up Discussion**
> >
> > Dear Reviewer eX2J,
> >
> > We greatly appreciate the time and effort you've invested in providing your insightful comments on our paper. We have carefully considered your feedback and have made corresponding responses that we hope will adequately address your concerns.
> >
> > Our primary goal is to clarify the motivations behind, and the implementations of, our proposed modules. Should you have any further questions or concerns, please feel free to reach out to us at your earliest convenience.
> >
> > Best regards,

---

> > > ### Comment · Reviewer_eX2J · 2023-11-23
> > > **Follow-up Discussion**
> > >
> > > Dear Authors,
> > >
> > > Thanks for preparing the rebuttal. I agree with Reviewer 1xNe that visual results for the ablation study are required so that the effectiveness of the proposed modules can be better visualized.
> > >
> > > Therefore, I would keep my original rating.
> > >
> > > Thanks,

---

### Official Review · Reviewer_1xNe · 2023-11-01

**Soundness:** 3 good
**Presentation:** 3 good
**Contribution:** 3 good
**Rating:** 6
**Confidence:** 4

**Summary:**

This paper proposed a versatile video generation model named VersVideo, which enhances spatial-temporal capability using textual, visual, and stylistic conditions. The model incorporates multi-excitation paths for spatial-temporal convolutions and multi-expert spatial-temporal attention blocks to improve temporal consistency and generation performance without escalating costs. It also uses temporal modules in the decoder to maintain inter-frame consistency and mitigate information loss. Besides, they design a unified ControlNet model suitable for various visual conditions.

**Strengths:**

- This paper is well organized. The authors have well categorized and summarized previous methods. The authors explain the shortcomings of previous methods and the motivation for proposing the method very clearly.
- The authors provide web demos for visualization. It is helpful for the readers to examine the effectiveness of the proposed method.
- The authors provide a specially designed module for perceiving temporal information for video generation tasks. The design of the module is very innovative, and it shows improvement in quantitative indicators.
- The decoder design and consistency loss are helpful for maintaining temporal consistency.

**Weaknesses:**

- The authors did not provide a visual comparison of the results. Providing a visual comparison would better demonstrate the effectiveness of the method. Besides, a user study is also needed to verify the effectiveness of the method.
- Adding the impact of each module on visual effects in the ablation experiments would be more helpful in verifying what role each module plays.
- The quality of the video-to-video editing results currently shown in the paper is not much better compared to other methods such as Render-A-Video and TokenFlow. The results generated in the current demo have severe color changes. Therefore, I doubt the effectiveness of this method in video fidelity.
- Since many modules in the framework designed in the paper are related to temporal consistency, please provide indicators to measure temporal consistency. Currently, the paper only lists indicators of generation quality.

**Questions:**

Can the current framework handle the situation with multiple condition inputs?

---

> ### Author Response · Authors · 2023-11-20
> **Response to Reviewer 1xNe [1/2]**
>
> Reviewer **1xNe** expressed a primary concern regarding the visual effects of our proposed model. In response, we made the folllowing revisions:
> 1. Added more text2video visual results in comparison to other prior models.
> 2. Conducted a user study to verify the effectiveness of *VersVideo*.
> 3. Demonstrated the potential of multiple control conditions for video generation.
>
>
> > Q1: The authors did not provide a visual comparison of the results. Providing a visual comparison would better demonstrate the effectiveness of the method. Besides, a user study is also needed to verify the effectiveness of the method.
>
> Thank you for your valuable feedback.
>
> We incorporate more visual examples of competing methods for comparison in our demo, which can be found at [link](https://docs.google.com/document/u/1/d/e/2PACX-1vTpTofxtnoetQwgFcVaHWx01Offc2sc4T5zRwY6I-tueiECmpZKbnrYqRQ9pXTdXsbvJelg2Y6OaW4n/pub).  Additionally, we provide a comparison of conditioned generation at the end of [demo](https://anonymous-pages.github.io/video_demos/).
>
> To conduct the user study, we selected four widely acknowledged and open-sourced models, namely Text2video-zero (Khachatryanet al., 2023), VideoFusion (Luo et al., 2023), Zeroscope (Hugging Face, 2023), and VideoComposer (Wang et al., 2023d), for visual comparison. The evaluation set consisted of 30 video prompts, and 16 evaluators compared the inter-frame consistency and overall video quality (including inter-frame consistency, alignment of text-video, and perceptual factors) between two videos - one from competing methods and one from our method, shown in a random sequence. Figure 11 shows that our method, *VersVideo*, was preferred over VideoFusion $83.3\\%$ of the time in terms of video quality, and over VideoComposer $88.5\\%$ of the time. Our method also performed significantly better than the baseline methods in terms of inter-frame consistency in the user study.
>
> For detailed results, please refer to [**Section E in the Appendix**](https://drive.google.com/file/d/1N0yYBc__UTsc6sABkA3FPkN0VOINHWpP/view?usp=sharing).
>
>
> > Q2: Adding the impact of each module on visual effects in the ablation experiments would be more helpful in verifying what role each module plays.
>
>
> Thank you for your insightful comments and suggestions.
>
> There are two fundamental modules designed for denoising UNet, namely, MEC and MOE-STA, as illustrated in Table 2. These modules work collaboratively to enhance the spatial-temporal quality of the generated videos. However, it's challenging to dissect their individual contributions to the final results, despite our attempts to visualize the videos from these ablation studies. The diffusion model is a complex system that intertwines all modules, making it difficult to distinguish the impact of each module through qualitative evaluation. Therefore, we rely on quantitative metrics to reveal their subtle effects, as shown in Table 2.
>
> Regarding TCM-VAE, we have conducted an ablation study presented in Table 3, which covers both video generation and video reconstruction using quantitative metrics. In addition, we provide visual demonstrations for [video reconstruction](https://anonymous-pages.github.io/video_demos/) in the second part of the demo and [video generation](https://docs.google.com/document/d/e/2PACX-1vQyICnc6WH8CgCCRuX0avsOI7MoO4gyPRQEro-v1UGu0nI60CTCiDksLUXOV5y_A597X41rL5mQdyER/pub) for different video diffusion models.

---

> ### Author Response · Authors · 2023-11-20
> **Response to Reviewer 1xNe [2/2]**
>
> >Q3: The quality of the video-to-video editing results currently shown in the paper is not much better compared to other methods such as Render-A-Video and TokenFlow. The results generated in the current demo have severe color changes. Therefore, I doubt the effectiveness of this method in video fidelity.
>
> Thank you for your valuable feedbacks.
>
>
> In our demo, the first example of video-to-video editing showcased severe color changes due to the text prompt ***"cyberpunk city"* which is associated with the metaverse, characterized by rapid color changes and flashing lights.**
>
> There are two concurrent studies, Render-A-Video and TokenFlow, for video editing. The Render-A-Video is a video-to-video model that employs multiple pipelines. It leverages the off-the-shelf image model from [CivitAI](https://civitai.com/), a public platform that enables artists to share their personalized models. In addition, TokenFlow utilizes pre-trained image models and the PnP image diffusion model [ref-1] for video editing.
>
> It would be reasonable to expect that their rendering results are more promising in terms of fidelity. This is because the video model is based on pretrained image models that were trained using high-quality images. However, there is a significant difference between their approach and the proposed *VersVideo* method. While Render-A-Video and TokenFlow are specialized for video-to-video applications, our model is designed to be versatile, allowing for a wider range of applications such as text-to-video, image-to-video conversion, and stylization.
>
>
> *Reference*:
>
> [ref-1] Tumanyan, N., Geyer, M., Bagon, S. and Dekel, T., 2023. Plug-and-play diffusion features for text-driven image-to-image translation. In Proceedings of the IEEE/CVF Conference on Computer Vision and Pattern Recognition (pp. 1921-1930).
>
>
>
>
> >Q4: Since many modules in the framework designed in the paper are related to temporal consistency, please provide indicators to measure temporal consistency. Currently, the paper only lists indicators of generation quality.
>
>
> Thank you for your thoughtful comments.
>
> We use the widely accepted Frechet Video Distance (FVD) [ref-1] and CLIP Similarity (CLIPSIM) [ref-2] as evaluation metrics for text-to-video generation. CLIPSIM assesses video quality and text alignment by averaging the CLIP similarity score of all generated frames. The FVD metric evaluates the similarity between real and generated videos, considering **both video quality and motion dynamics**. Therefore, FVD can serve as an indicator of temporal consistency. As most prior studies report FVD and CLIPSIM as the gold standard, we follow this practice to ensure a fair comparison.
>
>
> *Reference*:
>
> [ref-1] Unterthiner, T., Van Steenkiste, S., Kurach, K., Marinier, R., Michalski, M. and Gelly, S., 2018. Towards accurate generative models of video: A new metric & challenges. arXiv preprint arXiv:1812.01717.
>
> [ref-2] Wu, C., Huang, L., Zhang, Q., Li, B., Ji, L., Yang, F., Sapiro, G. and Duan, N., 2021. Godiva: Generating open-domain videos from natural descriptions. arXiv preprint arXiv:2104.14806.
>
>
> >Q5: Can the current framework handle the situation with multiple condition inputs?
>
> Thank you for your insightful comments.  Our model can handle multiple condition inputs.
>
> The global condition, such as the text prompt and style, is inherently compatible with other conditions. For instance, the final two rows in Fig. 6 utilize depth maps for motion structure control, while simultaneously employing images for global style control.
>
> When it comes to the composition of multiple motion structures, such as the concurrent use of depth, canny, and HED, we offer several examples to demonstrate its practicality with the [visual examples](https://docs.google.com/document/d/e/2PACX-1vQTN8K7LDRgAEAEniaQYD2ePbogQ__Hht7ewRqlzh0RNCfI13bIEPPzZwl67AK-gluB-UsYBagLd-R0/pub).

---

> > ### Comment · Reviewer_1xNe · 2023-11-22
> > **Response to Authors & Decision**
> >
> > Thanks for the authors' reply. After reading the authors' responses, I decide to keep my previous score unchanged. At the same time, I would not be surprised if this paper is rejected. The reasons are listed below:
> >
> > *Disadvantages*:
> >
> > 1. The author did not provide any change and analysis of the visual effects caused by each module in the ablation study, which is similar to the viewpoint of **reviewer eX2J**. The author only listed the impact of the final version of the VAE encoder on visual effects in the response. This makes me question the effectiveness of each module (especially on the visual effect).
> >
> > 2. The authors claim that the proposed module can enhance spatio-temporal consistency, but there is no corresponding metrics to prove it. FVD and average CLIP score cannot measure from the above dimensions. I suggest using metrics such as Frame consistency and motion control metric in *VideoComposer* to measure frame consistency. In addition, the temporal error in this [link](https://github.com/phoenix104104/fast_blind_video_consistency#evaluation) can also be used.
> >
> > 3. The authors did not explain how their model manages to handle multiple conditions simultaneously.
> >
> > *Advantages*:
> >
> > This article is indeed well-organized. The proposed module is technically novel. The authors have partially demonstrated its effectiveness.

---

> ### Author Response · Authors · 2023-11-22
> **Thank you for your feedbacks.**
>
> We appreciate your insightful feedback. We are sorry that our responses do not fully address your concerns.
>
> 1. We strongly agree that decomposing each module's visual effect for the final results would greatly help to gain an intuitive understanding of their contributions. In the paper, we rely on the quantitative metric to reveal these improvements. Admittedly, this could be our limitation of the response.
> 2. We are grateful for your suggestion regarding the frame consistency metric. We will add this metric shortly in the paper.
> 3. In the visual example, we generate videos with depth, HED, and canny collaboratively.  Our model is capable of handling multiple motion structure conditions because we train separate adapters for each condition,  and the features of these conditions can be added together for generation. This way of handling multiple conditions aligns with the application of multiple ControlNets (Zhang et al., 2023) or multiple T2I Adapters (Mou et al., 2023).

---

### Author Response · Authors · 2023-11-20
**Summary of Revision**

We would like to express our gratitude to all reviewers for their commitment to reviewing our paper. Their constructive and insightful comments have significantly improved our study. In response to the feedback, we provide individual responses to address each reviewer’s concerns, and an updated manuscript.
- Reviewer **1xNe** primarily expressed concerns about the visual quality of our models. In response, we have included more comparison examples with competing methods and, most importantly, added a user study as an additional validation metric.
- Reviewer **eX2J** raised substantial concerns about the motivation and details of our methodology. We acknowledge some missing details about the MoE spatial-temporal transformer, and the temporal compensated VAE for video decoding. To address this, we have provided supplementary explanations in Sections B, C, and D in the Appendix for better understanding.
- Reviewer **CWze** offered insightful perspectives concerning the limitations of our model. We have added a new section to discuss these limitations and the future scope of our study, as well as the applicability of our design to other video diffusion models.
- Reviewer **Cov9** questioned our contribution and the performance of our model compared with prior studies. In response, we have clarified our contributions in three aspects and also acknowledged related works. We have analyzed the performance our model achieves and demonstrated its competitiveness.


In summary, the core idea of our study is to build a versatile video generation model by developing an enhanced spatial-temporal denoising UNet, a temporal compensated VAE for video decoding, and a unified control strategy for versatile input conditions. We appreciate all reviewers for their time and feedback, and we hope that our changes adequately address all concerns.

---

### Meta-Review · Area_Chair_J12k · 2023-12-10

**Metareview:**

Three out of four reviewers placed the paper slightly above the borderline. The fourth reviewer suggested that the work is not good enough for acceptance. In their review, they asked several reasonable questions, to which authors responded in their clarification. The reviewer then suggested that they would like to keep the score since the authors didn't provide visual ablation results. The authors have provided numerical ablations instead. While the AC agrees that such results would be desired and recommends the authors to add them, the AC also thinks that this cannot serve as a grounds for rejection, especially for the "not good enough" rating. The other reviewers are more favorable to the manuscript. Hence, while seeing merit in the negative review, the AC thinks that the paper is a solid piece of work and hence recommends acceptance. Congrats!

**Justification For Why Not Higher Score:**

This is a borderline paper, the AC believe that poster is a high as it can go.

**Justification For Why Not Lower Score:**

We can drop the score lower here.

---

### Decision · Program_Chairs · 2024-01-16

Accept (poster)